# Task Vectors in In-Context Learning: Emergence, Formation, and Benefits

**Liu Yang**[w], **Ziqian Lin**[w], **Kangwook Lee**[w], **Dimitris Papailiopoulos**[w,m] **& Robert D. Nowak**[w]
[w]University of Wisconsin-Madison, [m]Microsoft Research
{liu.yang, zlin284, kangwook.lee}@wisc.edu,
dimitris@papail.io, rdnowak@wisc.edu

## Abstract

In-context learning is a remarkable capability of transformers, referring to their ability to adapt to specific tasks based on a short history or context. Previous research has found that task-specific information is locally encoded within models, though their emergence and functionality remain unclear due to opaque pre-training processes. In this work, we investigate the formation of task vectors in a controlled setting, using models trained from scratch on synthetic datasets. Our findings confirm that task vectors naturally emerge under certain conditions, but the tasks may be relatively weakly and/or non-locally encoded within the model. To promote strong task vectors encoded at a prescribed location within the model, we propose an auxiliary training mechanism based on a *task vector prompting loss (TVP-loss)*. This method eliminates the need to search for task-correlated encodings within the trained model and demonstrably improves robustness and generalization.[1]

## 1 Introduction

To understand the underlying mechanisms of in-context learning in transformers, researchers have probed pre-trained models from various perspectives, such as altering the labels in demonstrations (Min et al., 2022; Kim et al., 2022) and investigating circuit mechanisms (Elhage et al., 2021; Wang et al., 2023; Hanna et al., 2024; Singh et al., 2024). Additionally, controlled, small-scale studies have been conducted by training transformers from scratch to observe their in-context learning behavior on linear regression tasks (Garg et al., 2022; von Oswald et al., 2022; Lin & Lee, 2024), discrete functions (Bhattamishra et al., 2023), hidden Markov chains (Xie et al., 2021), and DFAs (Akyürek et al., 2024). Furthermore, theoretical approaches have also been applied to this problem (Xie et al., 2021; Lin & Lee, 2024; Giannou et al., 2023).

Among the various efforts to probe pre-trained models, one significant line of research employs the concept of a "task vector", which is a vector in the model's weight or activation space[2] that encodes task-specific information. The concept of the task vector was first introduced by Ilharco et al. (2022), where it is defined as a direction in a model's weight space corresponding to a particular task. Subsequently, Hendel et al. (2023) demonstrated that, given a task demonstration as context, a pre-trained large language model forms a task vector in its activation space at certain layers. This task vector encodes only the task information and is independent of the specific demonstration of the task. By inserting the task vector directly into the model, it is able to perform the task without context or demonstration (i.e., zero-shot). We will refer to this as *Task Vector Prompting* (TVP) in this paper. Concurrently, Liu et al. (2023a); Todd et al. (2023); Merullo et al. (2023a); Li et al. (2024b); Saglam et al. have also identified a single vector that encodes the task information, albeit using different terminology. We omit the details here and refer the reader to the related work and the original papers for more information.

Motivated by prior observations of task vectors in pre-trained LLMs—where training conditions are inherently difficult to control—we investigate their emergence in a controlled

---

[1]the code is available at https://github.com/Leiay/task_vector_prompting.
[2]The "activation space" refers to the space where the output of each transformer layer resides.

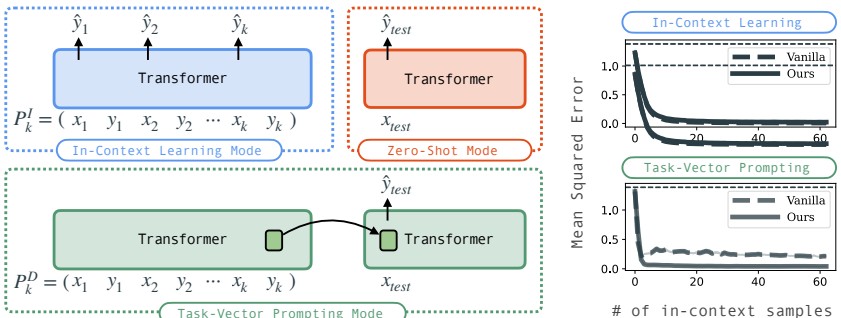

Figure 1: *Overview of the transformer operating in in-context learning (ICL) and task vector prompting (TVP) modes.* A transformer can be configured to operate in ICL mode, using input-output pairs as prompts, or in TVP mode, extracting task-specific embeddings for zero-shot predictions (architecture and training details in Section 3). In the bottom, the ICL and TVP performances of the vanilla-trained model and our method are shown, with the dashed horizontal line indicating random prediction performance (i.e., no task information is inferred). Compared to vanilla training, our approach enhances task-specific representations in the TVP mode while preserving comparable ICL performance.

setting by training transformers from scratch on synthetic datasets. As shown in Figure 1, when trained from scratch on the linear regression task defined as $y_i = \boldsymbol{w}^T \boldsymbol{x}_i$, the transformer (dashed line) demonstrates the ability to use in-context learning (ICL) to solve the task (bottom-left). We evaluate the trained model's performance in task vector prompting (TVP) mode (Figure 1, middle), where the task vector is extracted from the in-context learning mode and injected back in a zero-shot manner, as defined by Hendel et al. (2023). The TVP performance is much better than chance (indicated by the horizontal dashed line), but a bit worse than ICL performance, which we attribute to the fact that the encoding of the task may not be strong and localized by normal training methods. To encourage the formation of a strong and localized task vector, we propose an auxiliary training loss, called the *task vector prompting loss (TVP-loss)*. In our new approach, the model is trained using the TVP-loss in addition to normal training losses. As illustrated in Figure 1, our approach (solid line) achieves comparable ICL performance to the vanilla model in the ICL mode (bottom-left) and its TVP performance (bottom-right) is significantly improved and comparable to ICL performance with multi-shot demonstration.

Our main contributions and findings are outlined below:

**Emergence of Task Vectors During Training.** We investigate the effectiveness of task vector extraction methods–originally proposed for pre-trained LLMs–when applied to small-scale models trained from scratch on synthetic datasets. Our findings show that task vectors can naturally emerge during training, consistent with the observations in Hendel et al. (2023) for pre-trained LLMs, provided appropriate input formats and sufficient model capacity. To better understand what promotes this emergence, we examine various hyperparameters, including model depth, sparse attention, and input embedding size.

**Strong Task Vectors with Proposed Auxiliary Loss.** Although task vectors naturally emerge, they are often weak and entangled with information from input queries. This prevents task vectors from representing purely task-specific knowledge. To overcome this limitation, we propose a training algorithm that explicitly encourages the formation of task vectors independent of query information. This approach strengthens the task vector, ensuring that task-specific encoding is explicitly established within the model.

**Task Vectors for In-Context Learning Robustness.** Using our proposed training algorithm, we analyze the effects of TVP-loss on (a) synthetic, (b) formal language, and (c) natural language tasks, demonstrating enhanced robustness in in-context learning.

## 2 Related Works

The notion of a "task vector" was first introduced by Ilharco et al. (2022). Subsequently, Hendel et al. (2023), Merullo et al. (2023a), Yu et al. (2023), Liu et al. (2023a), Saglam et al. and Li et al. (2024b) demonstrated that a single vector in the model's activation space can encode learned functions in a pretrained model. Specifically, Liu et al. (2023a) showed that, when presented with an in-context learning demonstration, the activation at each layer points to a subspace encoding the task information. Additionally, Todd et al. (2023) identified the "function vector," another form of task vector, by averaging the causal attention heads, which can also guide the pre-trained language model's performance towards desired tasks.

Task vectors have also been identified under different modality (Luo et al., 2024; Hojel et al., 2024; Peng et al., 2024), and from different perspective, such as cognitive science (Piantadosi et al., 2024), and through theory perspective (Tao et al., 2024). Park et al. (2023) observe that concepts are represented linearly as directions in some representation space.

A recent study by Mittal et al. (2024) explores a setting similar to ours, examining whether learning appropriate latent representations via a bottleneck architecture enhances robustness in in-context learning. Notably, our approach facilitates the natural emergence of latent representations within the model's architecture. Expanding on this, Kobayashi et al. (2024) demonstrate that bottleneck architectures improve compositional generalization, though their definition of compositionality pertains to meta-skill learning rather than the sequential compositionality studied in our work. Additionally, Elmoznino et al. (2024) leverage bottleneck architectures to analyze prequential ICL performance. Furthermore, Han et al. (2024) investigate the dynamics of task vector formation in both sparse linear regression and pre-trained LLMs, showing that as task vectors emerge, models simultaneously develop conditional decoding strategies, leading to improved in-context learning performance.

Due to space constraints, extended related work appears in Appendix A.

## 3 Task Vector Definition

In this section, we formally define the notion of task vector. Let $\mathcal{F}$ denote a class of functions or "tasks", and let $\mathcal{X}$ and $\mathcal{Y}$ be the input and output spaces, respectively. If $f \in \mathcal{F}$ is the task in a specific "context", then for any input $x \in \mathcal{X}$, the corresponding output is $y = f(x) \in \mathcal{Y}$. Consider $x_{\text{test}} \in \mathcal{X}$, and the transformer model $M$. We can measure the model's zero-shot performance by $\ell(f(x_{\text{test}}), M(P_{\text{query}}))$, where $P_{\text{query}} = [x_{\text{test}}]$ denotes the query input. The in-context learning performance with $k$-shot examples is measured by $\ell(f(x_{\text{test}}), M(P_k))$, where $P_k$ is the in-context prompt. The prompt contains the $k$-shot examples $[x_1, f(x_1), \cdots, x_k, f(x_k), x_{\text{test}}]$.

Consider a *demonstrated prompt* $P_k^D$, where the superscript $D$ highlights its role in demonstrating task information. This prompt contains $k$ in-context samples: $P_k^D = [x_1, f(x_1), \cdots, x_k, f(x_k)]$ A task vector is an internal embedding $\tau$ extracted when the model is presented with $P_k^D$, which encodes the task at hand ($f$). Inserting the task vector into the model during zero-shot prompting is denoted by $M(P_{\text{query}}; \tau)$. Note that the task vector $\tau$ is extracted from the internal embeddings when inputting $P_k^D$, not $P_k$, therefore the task vector $\tau$ does not have explicit knowledge of $x_{\text{test}}$ when encoding the task.

A task vector extractor $g$ tries to locate this single embedding $\tau$ in the model given the demonstrated prompt $P_k^D$, i.e. $\tau = g(M(P_k^D))$. Then the performance of this extractor can be measured by $\ell(f(x_{\text{test}}), M(P_{\text{query}}; \tau))$. A task vector is considered successfully formed in the model $M$ for the task $f$ if the performance of the extracted task vector, i.e. task vector prompting performance, closely aligns with the in-context learning performance and outperforms zero-shot performance.

Hendel et al. (2023) confirmed that, in pre-trained large language models, $g(M(P_k^D))$ corresponds to the output embedding at approximately the middle layers of the model during the forward pass, which is responsible for predicting $f(x_{\text{test}})$. Similarly, Sia et al. (2024) found that the task in context is recognized by the model during the middle stage of the forward pass.

## 4 Emergence of Task Vectors in Trained-from-Scratch Models

### 4.1 Experimental Setup

We study task vector localization using a GPT-2 decoder trained on randomly generated prompts for the following tasks (details in Appendix B.1):

1. *Linear Regression.* $f(x_i) = w^\top x_i$ with $x_i, w \in \mathbb{R}^d$. We set $d = 6$ in our experiments.

2. *Sinusoidal Regression.* $f(x_i) = \sin(0.5 \cdot w^\top x_i + b)$, where $b \sim \mathcal{N}(0, 1)$.

3. *Discrete Token Offset Prediction.* $f(x_i) = (a \times x_i + b) \bmod C$, where $a \in \{1, 2, 3\}$, $b \in \{0, 1, 2\}$ sampled uniformly, and $C$ as the vocabulary size (default: $C = 1000$). mod denotes the modulo (remainder) operation.

During training, prompts are generated on the fly. Specifically, for each prompt, a function $f \in \mathcal{F}$ is randomly sampled according to the function distribution described for each task above. Subsequently, input tokens $\{x_i\}_{i=1}^{k}$ are independently sampled from the corresponding input distribution. The function $f$ is then evaluated on these inputs to produce the target outputs, forming the in-context learning prompt $P_k = [x_1, f(x_1), \ldots, x_k, f(x_k), x_{\text{test}}]$. Let the distribution of such prompts be denoted as $\mathcal{P}$. The transformer $M$ parameterized by $\theta$ is then trained to minimize the following expected loss:

$$\min_{\theta} \mathbb{E}_{P_k \sim \mathcal{P}} \sum_{i=1}^{k} \ell(f(x_{i+1}), M_\theta(P_i)),$$

where $P_i = [x_1, f(x_1), \ldots, x_i, f(x_i), x_{i+1}]$ represents the prompt prefix containing $i$ in-context examples. The loss function $\ell(\cdot, \cdot)$ is defined as mean squared error for regression tasks and cross-entropy loss for discrete token tasks. In our experiments, we use GPT-2 model with an embedding size of 64, 4 attention heads, and 3 layers, trained with a maximum context length of 63 (i.e., $k = 63$) and without positional embeddings (NoPE). We use the Adam optimizer with a learning rate of 0.0001 and a batch size of 256, training for 300k iterations. With dynamically generated prompts, this corresponds to 76.8 million distinct prompts. All experiments are run on an NVIDIA GeForce RTX 3090.

We investigate various input formats and task vector extraction methods (details in Appendix C.1 and Appendix C.2) to identify conditions under which task encoding emerges in trained-from-scratch transformers. Based on this investigation, we focus on the following setup, where task vectors emerge most distinctly:

- *input format*: Prompts with $k$ in-context examples are generated as $P_k = [z, x_1, f(x_1), \ldots, x_k, f(x_k), x_{\text{test}}]$, where $z$ is a special token placed at the beginning of the prompt to serve as a placeholder for injecting task encoding during zero-shot task vector prompting. During training, the embedding of $z$ is treated as a learnable parameter that is shared across all prompts (details in Appendix C.1).

- *task vector extractor method*: inspired by Hendel et al. (2023), we locate the task vectors *at the activations of the token $z$ and $\{y_i\}$* in the input format mentioned above. Specifically, during task vector prompting, the embedding is copied into the corresponding position in a zero-shot model, and the zero-shot loss is measured. A lower zero-shot loss indicates that the embedding effectively encodes task-specific information (details in Appendix C.2).

As noted in Hendel et al. (2023), the extraction process involves identifying the layer that optimally encodes the task vector. The optimal layer index $l^\star$ is determined by evaluating task vectors extracted from each layer and selecting the one that minimizes the average loss $\ell$ across all $N$ demonstrated prompts. This procedure ensures that the selected layer provides the most effective task vector for predicting the test outputs. A formal description of this procedure is provided in Appendix C.3.

*Remark* 4.1 (Task Vector Location). The work of Hendel et al. (2023) employs an in-context learning prompt format structured as $P_k = [x_1, z, f(x_1), \cdots, x_k, z, f(x_k), x_{k+1}, z]$, where $z$ indicates the "maps-to" token. This setup differs from the prompt format used in our experiments. In Appendix C.1, we investigate various input formats to determine their effect on task vector emergence. We find that the trained-from-scratch model exhibits task vector emergence only when the input format for training is $P_k = [z, x_1, f(x_1), \cdots, x_k, f(x_k), x_{k+1}]$, where no additional tokens are placed between $x$ and $f(x)$.

### 4.2 Task Vector Emergence

**Finding 1:** *Task vectors naturally emerge in small models trained from scratch.*

In Figure 1, we demonstrate the presence of task encoding in the activations of token $y_i$ under the linear regression task, using the experimental setup described in Section 4.1

with number of demonstrated prompts $N = 50$. In the task vector prompting mode, the transformer achieves a mean squared error (MSE) of approximately 0.25, significantly lower than the random prediction baseline of around 1.2, indicating that the model successfully infers task information in its representation. Throughout the paper, we use $N = 50$ by default unless stated otherwise. Additionally, we examine the impact of different $N$ values on task vector prompting performance, as detailed in Appendix C.6.

In the following sections, we analyze the impact of model depth and context length on task vector emergence across different problem dimensions (Section 4.2.1) and investigate the locations of task vectors (Appendix C.3.1). Additionally, we explore the effects of embedding size and sparse attention in Appendices C.4 and C.5, respectively.

### 4.2.1 Effects of Model Depth and Context Length on Task Vector Emergence

In the linear regression task, we examine how model depth and context length affect the emergence of task vectors by varying the problem dimension and model depth. We explore $d \in \{4, 5, 6, 7, 8, 9\}$, keeping the transformer's embedding size at 64 and adjusting the model depth from $L = 3$ to $L = 8$. The transformer's performance in ICL mode (solid line) and TVP mode (line with triangular markers) is shown in Figure 2, with $N = 50$ for clarity.

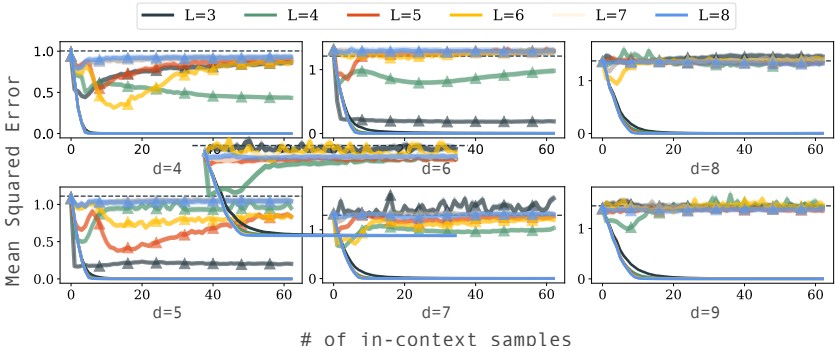

Figure 2: *Performance of the transformer in in-context learning (ICL) mode (solid line) and task vector prompting (TVP) mode (line with triangular markers) across problem dimensions $d \in \{4, 5, 6, 7, 8, 9\}$ and varying model depths $L \in \{3, 4, 5, 6, 7, 8\}$. The dashed line indicates the random-guess baseline. The results indicate that smaller problem dimensions ($d = 4$ to $d = 7$) and shallower model depths ($L = 3$ to $L = 5$) yield stronger and more stable task encoding in the TVP mode. However, task encoding remains noisy in most cases. Task vectors emerge more clearly when the ICL loss plateaus but tend to disperse with increasing in-context length. In deeper models, task information appears to distribute across layers, reducing the distinctiveness of task vectors.*

> **Finding 2:** *In the linear regression task, task vectors are most distinct with moderate model depth and just before the in-context learning loss plateaus.*

We observe the following characteristics of task vectors:

**Along the Model Depth:** Tasks with smaller problem dimensions ($d = 4$ to $d = 7$) exhibit a clear pattern of specific model depths producing strong and stable task encoding, as illustrated in the plot. However, in most cases, task encoding remains noisy. Notably, as model depth increases, the clarity of task encoding diminishes. This could be due to the model's increased capacity to approximate the least square solution directly in a single forward pass, reducing its dependence on previously calculated task encoding.

**Across the In-Context Length:** Task vectors become more distinct as the in-context learning loss plateaus, but they tend to disperse afterward. For example, when $d = 5$ with $L = 4$ or $d = 7$ with $L = 4, 5, 6$, task encoding is most evident around the 5-th in-context example but then disperses into random task encodings. This suggests that the model initially learns the task in a compact and focused manner; however, over time, the task information becomes distributed more broadly across the context.

# 5 A New Training Algorithm to Encourage the Formation of Task Vector

As evidenced in the previous section, it is difficult to locate a single vector in the trained-from-scratch small-scale transformer model that cleanly encodes the task at hand. In this section, we propose a training algorithm that explicitly encourages the formation of task vectors (as defined in Hendel et al. (2023)) in the in-context learning process.

Through this algorithm, which is a straightforward extension of the task vector definition, we obtain a model with the task vector explicitly formed. As demonstrated in Section 5.2, comparing this model to the vanilla model (where task vectors are not explicitly formed) reveals that task vector formation enhances the model's robustness in in-context learning tasks, particularly with out-of-distribution prompts. This highlights the value of task vector formation in improving generalization and interpretability in in-context learning.

## 5.1 A New Training Algorithm to Encourage Task Vector Formation

To encourage the formation of task vectors, we include the performance metric of the task vector in the training loss. Specifically, following the notation in Section 4, let each random prompt be $P_k = [z, x_1, f(x_1), \cdots, x_k, f(x_k), x_{k+1}]$ to be sampled following the prompt distribution $\mathcal{P}$, and a random test prompt $P_{\text{query}} = [z, x_{test}]$. We train the transformer $M_\theta$ by optimizing the following loss:

$$\min_\theta \mathbb{E}_{P \sim \mathcal{P}} \sum_{i=1}^{k} \left[ \underbrace{\ell(f(x_{i+1}), M_\theta(P_i))}_{\text{ICL-loss}} + \underbrace{\ell(f(x_{\text{test}}), M_\theta(P_{\text{query}}; h_i^l))}_{\text{TVP-loss}} \right], \tag{1}$$

where $h_i^l$ represents the hidden state at the $l$-th layer of the transformer, extracted from the $i$-th in-context example. Instead of adaptively identifying the layer where the task vector resides, we simplify the process by directly designating a specific layer $l$ as the location for forming the task vector. This ensures consistent representation of task-specific information and facilitates training. We illustrate the training algorithm in Figure 13. In Section 5.2, we further explore the impact of this hyper-parameter choice on the model's performance.

The loss comprises two complementary components: (1) **In-Context Learning Loss (ICL-loss)**: The term $\ell(f(x_{i+1}), M_\theta(P_i))$ trains the model to predict the output $f(x_{i+1})$ for the $(i+1)$-th example, based on the preceding context $P_i$. Summing over $k$ examples ensures the model learns from the entire context effectively. (2) **Task Vector Prompting Loss (TVP-loss)**: The term $\ell(f(x_{\text{test}}), M_\theta(P_{\text{query}}; \tau))$ with $\tau = h_i^l$ evaluates the model's ability to use the injected task vector $h_i^l$, derived from the hidden state at the $i$-th in-context example, to predict the test output $f(x_{\text{test}})$. This term encourages the model to encode task-specific information in the task vector $\tau$. As evidence of this, in Figure 1, although the model trained with ICL-loss already exhibits emergence of task vector, the model with TVP-loss further improves the encoded task vector, achieving a lower TVP error.

## 5.2 Experimental Results on Synthetic Tasks

We apply this training algorithm to the aforementioned three synthetic tasks: (1) the linear regression task, (2) the sinusoidal regression task, and (3) the discrete token offset task described in Section 4.1. For all experiments, the transformer model is configured with a total of 8 layers. For consistency to the previous results, we set $N = 50$.

> **Finding 3:** *The proposed TVP-loss enhances task vector prompting performance, aligning it closely with in-context learning performance.*

For clarity, we evaluate the in-context learning and task vector prompting performance at the 63rd context (i.e., the final query) for models trained with and without the proposed TVP-loss. These results remain consistent across various context lengths, with full details provided in Appendix D.2. In this experiment, we designate task vectors to form at the 1st, 3rd, 5th, and 7th layers. As illustrated in Figure 3, for the 8-layer transformer, the task

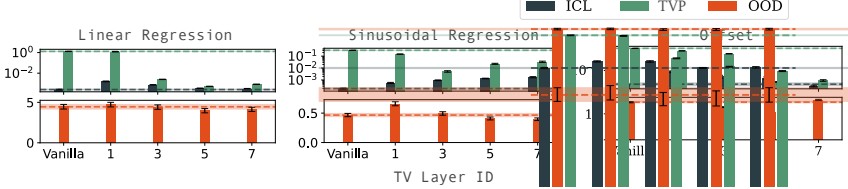

Figure 3: *Model performance with vanilla and TVP-loss training.* This figure compares models trained with vanilla training and TVP-loss across layers where task vectors are formed. The upper row shows in-context learning (ICL) and task vector prompting (TVP) performance on a logarithmic scale for clarity, while the bottom row depicts out-of-distribution (OOD) prompt performance. Dashed lines represent the performance of the vanilla-trained model in each respective setting (ICL, TVP, and OOD), providing baselines for comparison. Each column corresponds to a task: linear regression (left), sinusoidal regression (middle), and discrete token offset (right). The horizontal axis indicates the layer at which task vectors are formed. **ICL performance:** Models with TVP-loss achieve similar ICL performance to vanilla-trained models. **TVP performance:** TVP-loss improves TVP performance, especially when task vectors form in intermediate or later layers. **OOD robustness:** TVP-loss enhances OOD generalization, notably in complex tasks like sinusoidal regression and discrete token offset, with task vectors formed at specific layers.

vector prompting performance of the vanilla-trained model is nearly random. In contrast, models trained with the TVP-loss exhibit task vector prompting performance that closely matches their in-context learning performance, as long as the task vector layer is set to an intermediate or later layer rather than the initial layers of the model.

**Performance on OOD Prompts.** Forcing the task vector to form at a specific layer can be interpreted as introducing an implicit bottleneck architecture within the model's forward pass. In this section, we examine the model's generalization ability when the task vector is constrained to form at a particular layer. Specifically, for the three synthetic tasks, we evaluate performance on the OOD prompts described in Appendix B.1, with additional OOD tasks and their results are presented in Appendix D.3.

Note that during training, none of these OOD settings were encountered. As shown in Figure 3, models with task vectors formed at specific layers exhibit equal or improved performance on OOD prompts compared to vanilla-trained models. For the linear regression task, forming the task vector at the 5th and 7th layer yields slightly better OOD performance.

The sinusoidal regression task can be consider as a compositional task, where $f_1(x_i) = w^T x_i$, and $f_2(y) = \sin(y)$. Then the OOD task modifies the $f_2(y) = \sin(y)$ to $f_2(y) = \sqrt{y}$. In this case, models with task vectors show enhanced generalization to the modified compositional task, indicating that capturing task representations at intermediate layers improves OOD generalization. We provide an extended analysis of the compositional functions in Appendix D.4.

In the discrete token offset task, models with task vectors handle context changes significantly better than those trained without the auxiliary loss. This suggests that the auxiliary loss enables the model to adapt more effectively by filtering out unrelated context at earlier positions, compared to vanilla-trained models.

## 6  In-Context Learning Beyond Synthetic Tasks

> **Finding 4:** *TVP-loss improves in-context learning performance on out-of-distribution prompts.*

In this section, we extend our study beyond the simple synthetic tasks on regression and token offsets, and investigate the impact of the TVP-loss in synthetic formal language in Section 6.1 as well as the natural language tasks in Section 6.2.

### 6.1  Synthetic Formal Language Task

We conduct experiments on two benchmarks: the Generative In-Context Learning (GINC) dataset introduced in Xie et al. (2021), and the RegBench dataset from Akyürek et al. (2024). In both datasets, each context consists of $s$ tokens per example. For instance, an in-context prompt with context length $k = 3$ and example length $s = 4$ looks like: a b c - d / e f g - h / i j k - l, while a corresponding zero-shot prompt is m n o - p. Further experimental details are provided in Appendix B.2.

**GINC Dataset.** Though the model is not Meta-ICL trained—meaning the pre-training distribution differs from the prompt distribution—training with the TVP-loss still enables

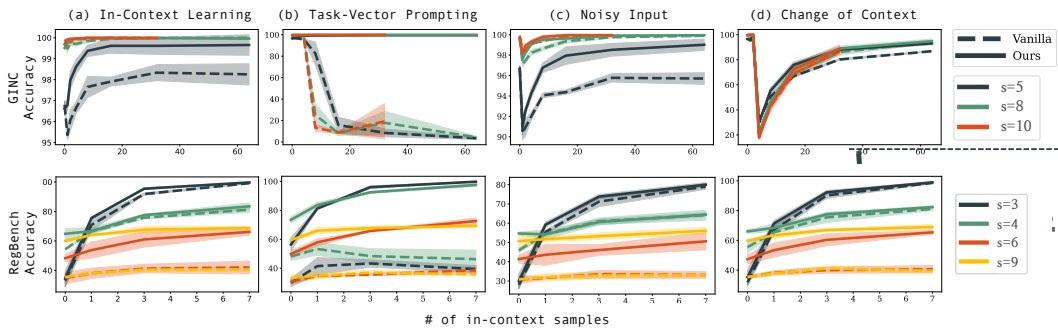

Figure 4: *GINC (top) and RegBench (bottom) dataset in-context learning performance.* Comparison of the in-context learning performance for the vanilla-trained model (dashed line) and the model trained with TVP-loss at the 5th layer for GINC and the 4th layer for RegBench (solid line). In all cases, the model with task vector formation outperforms the vanilla-trained model.

the formation of task vectors during in-context learning. We present the result in Figure 4 (top). We measure both model's performance on (a) in-context learning, (b) task vector prompting, (c) noisy in-context learning, where the in-context token is replaced with a random token with probability 0.1, and (d) when the underlying HMM is changed after the second context.

**RegBench Dataset.** In Figure 4 (bottom), we present results on: (a) in-context learning, (b) task vector prompting, (c) noisy in-context learning, where 4 out of 32 in-context labels are replaced with random tokens, and (d) performance when the underlying DFA of the first 4 out of 32 samples is changed to another DFA. When evaluating the next predicted token in the in-context learning prompt, we follow the same setup as in Akyürek et al. (2024): as long as the next predicted token belongs to the outgoing edge set of the current DFA state, it is considered a correct prediction; otherwise, it is incorrect. Intuitively, as $k$ increases, more transitions are presented to the transformer. This requires the model to approximate a larger $n$-gram distribution, leading to a degradation in performance.

**Performance Analysis.** Across both datasets, models trained with TVP-loss consistently outperform vanilla-trained models. Specifically, in the GINC dataset, TVP-loss improves the model's ability to learn in-context, even when trained only on next-token prediction tasks, and enables task vector formation without directly optimizing for in-context learning during pre-training. As shown in Figure 4, models trained with TVP-loss achieve slight improvements in ICL performance and significant gains in TVP performance. Additionally, these models exhibit increased robustness to label noise and out-of-distribution (OOD) in-context samples, consistently outperforming vanilla-trained models in OOD scenarios.

## 6.2 Natural Language Task

Motivated by the OOD improvements observed with TVP-loss on GINC and RegBench, we further assess its effectiveness in natural language tasks using the CrossFit benchmark (Ye et al., 2021) following the MetaICL settings in Min et al. (2021). Specifically, we fine-tune a pretrained GPT-2 Large model (774M parameters), modifying the original MetaICL procedure by training across all context lengths rather than using a fixed context length ($k = 16$), which allows the model to generalize to varying context lengths at inference time.

Since the meta-training set contains a diverse range of tasks, the optimal layer for forming task vectors may vary across tasks. To accommodate this, we do not fix a single task vector layer during fine-tuning with TVP-loss. Instead, for each training prompt, we evaluate the TVP-loss at every layer from 8 to 15 (out of 36 total layers) and select the layer that minimizes the loss. Formally, recall the TVP-loss in equation (1) for layer $l$ is defined as

$$\ell_l^{\text{TVP}} = \sum_{i=1}^{k} \ell(f(x_{\text{test}}), M_\theta(P_{\text{query}}; h_i^l)).$$

For each training step and each prompt, instead of minimizing $\ell_l^{\text{TVP}}$ with predefined $l$, we minimize $\ell^{\text{TVP}} = \min_{l \in \{8,9,\dots,15\}} \ell_l^{\text{TVP}}$. This allows the model to determine the most effective task vector location per prompt.

During inference, following Min et al. (2021), each test query is paired with a set of candidates (e.g., classification labels or answer choices for QA), and the model selects the candidate with the highest conditional probability.

Table 1 presents results under three task transfer settings in the CrossFit benchmark following MetaICL settings. In each case, we train on a set of meta-training tasks and evaluate on a distinct set of target tasks to assess generalization. The three settings are:

1. High-resource → Low-resource (HR→LR): Datasets with 10,000 or more training examples—referred to as high-resource tasks—are used for meta-training, while the remaining tasks are used for evaluation.

2. Non-Class → Class: The model is trained on tasks that do not involve classification and then evaluated on classification tasks.

3. Non-Paraphrase → Paraphrase: The model is trained on tasks not with paraphrase detection and evaluated on tasks with paraphrase detection.

We report in-context learning performance (with context length $k = 16$) on the target tasks, showing the mean and standard deviation over 5 inference seeds. In addition to overall target task performance, we also separately evaluate on unseen domain target tasks–datasets whose topics (e.g., finance, poetry, climate, or medical) do not appear during meta-training– to assess cross-domain generalization. As shown in Table 1, models meta-trained with TVP-loss consistently outperform those without TVP-loss. We hypothesize that TVP-loss serves as a regularizer, promoting better OOD generalization by encouraging the model to form task-specific encoding.

Lastly, we evaluate the performance of task vector prompting with $N = 10$ demonstrations. For each test query, we follow the same inference strategy as in standard in-context learning: pairing the query with a set of candidate answers. Instead of passing the full demonstration context, we insert the task vector extracted from the context into a specific layer determined by the $N$ demonstrations. The task vector prompting results are reported in the lower half of Table 1. Notably, although the evaluation tasks were never seen during training, TVP-loss still provides a modest performance gain in HR→LR and Non-Paraphrase→Paraphrase settings, suggesting that it also improves task localization ability.

| | Method | HR→LR | non-Class→Class | non-Paraphrase→Paraphrase |
|---|---|---|---|---|
| ICL | MetaICL | $44.88 \pm 0.29$ / $39.14 \pm 3.20$ | $37.97 \pm 1.05$ / $32.71 \pm 1.51$ | $41.07 \pm 2.57$ / $34.05 \pm 0.00$ |
| | + TVP-loss (Ours) | $\mathbf{45.60} \pm 0.61$ / $\mathbf{42.10} \pm 3.67$ | $\mathbf{39.55} \pm 1.05$ / $\mathbf{33.78} \pm 2.43$ | $\mathbf{41.82} \pm 1.73$ / $34.05 \pm 0.00$ |
| TVP | MetaICL | $31.93 \pm 0.20$ | $\mathbf{27.74} \pm 0.34$ | $30.97 \pm 2.00$ |
| | + TVP-loss (Ours) | $\mathbf{34.41} \pm 0.25$ | $22.71 \pm 1.56$ | $\mathbf{33.45} \pm 0.82$ |

Table 1: *ICL and TVP Accuracy of Fine-tuned GPT-2 Large on the CrossFit Benchmark.* We evaluate performance under three task transfer settings: HR→LR (High-Resource to Low-Resource), non-Class→Class, and non-Paraphrase→Paraphrase. Each cell reports the mean ICL accuracy ($\pm$ std) over three inference seeds using $k = 16$ in-context examples, shown as: overall performance / performance on target tasks from unseen domains. Bold values indicate that the difference between MetaICL and ours exceeds the sum of their standard deviations.

## 6.3 Effects of Various Hyperparameter Choices

To further understand the robustness and sensitivity of our approach, we analyze the impact of key hyperparameters on the HR→LR task:

(A) the range of candidate layers used for selecting the task vector location, and

(B) the weight $w_{\text{TVP}}$ in the combined objective $L_{\text{ICL}} + w_{\text{TVP}} \cdot L_{\text{TVP}}$ (see Equation 1).

Table 2 reports ICL and TVP performance under different configurations.

**Layer range:** Expanding the candidate range from the default (8–15) to a broader span (8–18) generally improves ICL performance but degrades TVP accuracy, potentially due to noisier task vector selection when the range is too wide. Conversely, narrowing or shifting the range (8–12, 6–13, or 10–17) slightly degrades both ICL and TVP performance. These results suggest that the 8–15 range strikes a balance between flexibility and task vector quality.

**Loss weight $w_{\text{TVP}}$:** Varying the weight of the TVP-loss in the joint objective reveals a trade-off: smaller weights (e.g., $w_{\text{TVP}} = 0.3$) slightly improve ICL performance but reduce TVP

accuracy, whereas larger weights (e.g., $w_{\text{TVP}} = 3$) hurt both metrics—likely due to over-regularization. We find that $w_{\text{TVP}} = 1$ yields the best overall balance between in-context learning and task vector generalization.

| Range | 8-15 | 8-12 | 8-18 | 6-13 | 10-17 | 8-15 | 8-15 |
| $w_{\text{TVP}}$ | 1 | 1 | 1 | 1 | 1 | 0.3 | 3 |
| ICL | $45.60 \pm 0.61$ | $45.33 \pm 0.30$ | $46.00 \pm 0.09$ | $45.45 \pm 0.52$ | $45.60 \pm 0.47$ | $45.93 \pm 0.45$ | $44.53 \pm 0.40$ |
| TVP | $34.41 \pm 0.25$ | $34.62 \pm 0.33$ | $32.91 \pm 0.55$ | $33.99 \pm 0.22$ | $33.28 \pm 0.68$ | $33.83 \pm 0.10$ | $33.65 \pm 0.11$ |

Table 2: *ICL and TVP Accuracy of HR→LR Task under Various Hyperparameter Setup.*

## 7 Discussion and Conclusions

**Factors Affecting Task Vector Emergence.** The emergence of the task vector in pre-trained large language models may be attributed to multiple factors, such as the model's capacity, scaling laws, and the diverse tasks encountered during pre-training. While Hendel et al. (2023) and related studies identify and analyze the existence of task vectors, they do not study the factors that may effect their emergence and performance. This gap in understanding motivates our investigation: we reproduce this phenomenon in the small-scale setting when trained from scratch, gaining insight into the factors influencing task vector emergence, such as input format, model depth, and context length. For instance, in the linear regression task, when the model is trained with prompts where $x$ and $y$ alternate closely, a model with moderate depth encourages the emergence of task vectors, which is most evident before the in-context learning loss plateaus.

**Training with TVP-Loss.** While task vectors can emerge through normal training processes, adding the TVP-loss to the training process encourages the formation of task vectors at prescribed locations and leads to improved accuracy and robustness. Across various benchmark datasets, we have demonstrated that prescribing the task vector to an intermediate or later layer of the model results in comparable in-context learning performance while forming strong task vectors. Depending on the task, task vectors formed at specific layers can enhance the model's robustness to out-of-distribution (OOD) prompts, most likely at the intermediate or later layer.

**Potential Applications of Task Vectors** Task vectors can serve as a soft prompt, effectively compressing the entire context into a single vector representation. In practical scenarios, where the number and nature of tasks are unknown during pre-training and only demonstrations are provided to the model, task vectors enable both context summarization and task identification. Specifically, a model with task vectors formed gains the ability to identify the underlying task from demonstrations, cluster tasks using the extracted task vectors, and subsequently perform zero-shot inference with these extracted vectors.

Another practical benefit is during inference, attention computations over earlier tokens can be masked out after the prescribed $l$-th layer, restricting attention to only the task vector and the query. This approach mirrors the findings of Sia et al. (2024), who demonstrated a 45% reduction in computation for pre-trained LLMs. Looking ahead, incorporating the TVP-loss as an auxiliary objective during training offers a promising approach to enhance task-specific representation and overall performance.

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

# A  Related Work (cont.)

**Understanding Pre-trained Large Language Model's Depth-Dependent Behavior.** Researchers have extensively studied the layer- and depth-dependent behaviors of large language models (LLMs) along the forward pass. nostalgebraist (2020), Belrose et al. (2023), and Wendler et al. (2024) analyzed how GPT models gradually determine top tokens through the logit lens and embedding space (Dar et al., 2022; Geva et al., 2022b). Lioubashevski et al. (2024) further highlighted that transformers identify top-$k$ tokens sequentially across layers. Beyond token predictions, several works investigate layer-specific encoding of information. For instance, Lv et al. (2024), Yu et al. (2023), Meng et al. (2022), and Gurnee & Tegmark (2023) explore how factual associations are stored in a layer-dependent manner. Additionally, Liu et al. (2023b) studied contextual sparsity changes across layers, while Sia et al. (2024) observed that "task recognition" typically occurs during the middle stages of the forward pass. More broadly, phase transitions in transformer behavior across layers have been discussed by Lad et al. (2024).

Leveraging these insights into layer-dependent behaviors, several works propose strategies to enhance LLM performance. Sharma et al. (2023) demonstrated that selectively applying low-rank decomposition in specific layers can improve reasoning ability in question-answering tasks. Meanwhile, Sia et al. (2024) and Liu et al. (2023b) introduced methods to reduce inference computation by identifying and leveraging specific layer behaviors. The concept of task vectors introduced in Hendel et al. (2023) also reflects a layer-dependent behavior in the context of in-context learning.

**Understanding Pre-trained Large Language Model's In-Context Learning Behavior** Since the launch of GPT-3 (Brown et al., 2020) and the observation that, during inference, the model can leverage few-shot examples to improve performance, researchers have been keenly interested in understanding the mechanisms of this in-context learning behavior. Several studies have explored the significance of labeling in context demonstrations (Min et al., 2022; Kim et al., 2022; Kossen et al., 2023; Long et al., 2024), while others have examined the phenomenon through circuit mechanisms (Elhage et al., 2021; Wang et al., 2022; Singh et al., 2024; Hanna et al., 2024; Wang et al., 2023). Research by Dai et al. (2022), Geva et al. (2022a), and Merullo et al. (2023b) links in-context learning behavior to implicit internal weight and activation updates within the model. To better understand "learning" in-context, several works (Pan et al., 2023; Sia et al., 2024) have studied its two distinct phases–task recognition and task learning–using controlled experiments designed to disentangle these phases. Park et al. (2024) studied the phase transition along the context length and depth. These studies focus on pre-trained large language models without access to the pre-training procedure. The literature on embedding editing and task vector is discussed in the main text.

**Understanding In-Context Learning Behavior in the Controlled Small-Scale Setting on Data-Fitting Problems.** Researchers have also examined in-context learning behavior in small-scale models, where models are trained from scratch to perform in-context learning tasks. Garg et al. (2022); Akyürek et al. (2022); von Oswald et al. (2022; 2023) investigated the

ability of transformers to learn regression problems, interpreting the model as performing a single step of gradient descent. This line of research was extended further in different pre-training mixtures of tasks (Raventos et al., 2023; Yadlowsky et al., 2023a), the ability to generalize to unseen tasks (Yadlowsky et al., 2023b), explanations via the Bayesian optimal estimator (Zhang et al., 2023; Bai et al., 2023; Muller et al., 2021), the viewpoint of generalization error (Li et al., 2023), and the perspectives of task retrieval and task learning (Lin & Lee, 2024; Wies et al., 2023).

Beyond regression, several studies have explored in-context learning abilities in other domains, such as reinforcement learning (Lee et al., 2023; Lin et al., 2023), discrete function learning (Bhattamishra et al., 2023), factorial hidden Markov chains (Xie et al., 2021), and deterministic finite automata (DFAs) (Akyürek et al., 2024). Swaminathan et al. (2023) studied the ICL mechanism with the clone-structured causal graphs to understand the schema learning, retrieval and rebinding. Additionally, Chan et al. (2022a;b) investigated the behavior of in-context learning and in-weight learning on the Omniglot datasets.

**Task Vector Applications**  Li et al. (2024a) utilized this task vector to enhance test-time adaptation, while Pham et al. (2024) employed it to erase malicious concepts during pre-training. Beyond activations, task information can also be encoded in a task token (Bai et al., 2024), or a pause token can be used to gain extra computation time (Goyal et al., 2023). Zhao et al. (2024) propose to present each task concept with a Gaussian distribution. Following the notion of "task vector" in the weight space, several works utilize the weight space difference for chatbot on new language (Huang et al., 2023; Kang et al., 2024), emotion transfer (Kalyan et al., 2024), aesthetic assessment (Yun & Choo, 2024), and soft prompt initialization (Belanec et al., 2024).

**Soft Prompt and Efficient Adaptation of Transformers.**  In this work, we primarily follow the definition of a task vector as described by Hendel et al. (2023). This task vector effectively compresses the information in the context into a single vector, which can be injected into the model to perform zero-shot inference. The notion of a task vector is also related to the concept of a soft prompt (Lester et al., 2021; He et al., 2022; Liu et al., 2021; Xu et al., 2023; Kang et al., 2024), where the context information is similarly compressed. Beyond learning soft prompts that encode instructions, researchers have also explored context compression by modifying the attention mask to ensure the summarization of context at a specific token (Ren et al., 2023; Mu et al., 2023; Phang, 2024). In the format of ICL, the researcher also study the continuous representation in context (Zhuang et al., 2024), as a reminiscent of the soft encoding.

Although we propose a training algorithm to encourage the formation of task vectors, our primary goal is not to compare our method of encoding task vectors with these approaches in soft prompt learning. Instead, we aim to understand the benefits of incorporating task vectors into in-context learning.

**Latent Representations and Sufficient Statistics in In-Context Learning.**  The TVP-loss in our approach can be interpreted as encouraging the model to form a latent representation that serves as a sufficient statistic of the task. Prior works have explored similar ideas in different contexts. As discussed in the main text, Hendel et al. (2023) observe that pre-trained LLMs first summarize the task into a latent vector representation, which is then used to guide task execution. Mittal et al. (2024) investigate whether learning an appropriate latent representation–via a bottleneck architecture–can improve robustness in in-context learning. Additionally, Kobayashi et al. (2024) show that such bottleneck architectures can enhance compositional generalization. These approaches are closely related to amortization-based meta-learning frameworks (Garnelo et al., 2018; Wu et al., 2025), and to methods that learn belief-state representations (Hu et al., 2024) or predictive state representations in reinforcement learning (Littman & Sutton, 2001; Ni et al., 2024). Our work connects to this line by explicitly supervising the formation of task-representative latent vectors through the proposed TVP-loss.

## B  Experimental Setup

### B.1  Synthetic Experiments

***Linear Regression.***  Following the setup in (Lin & Lee, 2024; Garg et al., 2022), the input vectors $x_i \in \mathbb{R}^d$ are sampled from a normal distribution $\mathcal{N}(0, I)$. The linear function weights $w \in \mathbb{R}^d$, which define the task, are sampled from a mixture of $d$ Gaussians, each with means corresponding to the standard basis vectors: $\mu_i \in \mathbb{R}^d$ with $\mu_{ij} = 1$ if $i = j$, and 0 otherwise. All Gaussians share a covariance matrix $\Sigma = \frac{1}{4} I$, and each component is selected with equal probability. The target outputs are computed as $f(x_i) = w^\top x_i$. We set $d = 6$ in our experiments.

***Sinusoidal Regression.***  Following the same sampling strategy as the linear regression task, the target outputs are defined as $f(x_i) = \sin(0.5 \cdot w^\top x_i + b)$, where $b \sim \mathcal{N}(0, 1)$. In this task, $w$ and $b$ determines the underlying function.

***Discrete Token Offset Prediction.***  In this task, we set the vocabulary size to be $C$, and each input token $x_i$ is a discrete value drawn from the set $\{0, 1, \ldots, C - 1\}$. The target output is defined as $f(x_i) = (a \times x_i + b) \mod C$, where $a \in \{1, 2, 3\}$, $b \in \{0, 1, 2\}$ are uniformly sampled, and mod denotes the modulo (remainder) operation. The task is uniquely determined by the choice of $a$ and $b$, resulting in total 9 possible tasks. By default, we set $C = 1000$.

For the three synthetic tasks, we evaluate the following out-of-distribution (OOD) tasks:

1. *Linear Regression*: We introduce an OOD prompt using the quadratic regression task, defined as $f(x_i) = w^\top(x_i \cdot x_i)$, where $\cdot$ denotes element-wise multiplication.
2. *Sinusoidal Regression*: we consider the OOD prompt where the task is $f(x_i) = \sqrt{\max(0.5 \cdot w^T x_i + h, 0)} + b$, where $h, b \sim \mathcal{N}(0, 1)$.
3. *Discrete Token Offset*: the OOD prompt switches the context at the 6th position.

### B.2  In-Context Learning for Formal Language

We provide the full experimental setup details for the two formal language datasets below.

#### B.2.1  The GINC Dataset: the Factorial Hidden Markov Chain.

The GINC dataset is introduced in Xie et al. (2021) to study the behavior of language pretraining and the emergence of in-context learning. We follow the setup as described in Xie et al. (2021): we define a uniform mixture of HMMs over a family of five concepts. Each entity and property is assigned unique tokens, resulting in a total vocabulary size of 100. To generate samples, we first select an HMM from the mixture and use it to produce documents containing 576 tokens. Next, we randomly insert 192 dummy tokens, denoted as "-," into these documents. These dummy tokens are labeled as -100 to be ignored during processing, resulting in documents with a total length of 768 tokens.

We use the GPT-2 model with 8 layers, 8 heads, and an embedding dimension of 256. The context length of this transformer is set to 768. To train the model to form a task vector at the $l$-th layer, we sample 10 extra tokens from the same HMM and randomly insert a dummy token "-". This forms the "zero-shot" document. The task vector loss is then calculated by injecting the $l$-th layer's hidden states at the token "-" into the "-" token in the zero-shot document. This training pipeline (shown in Figure 17) differs from the Meta-ICL pipeline described earlier (Figure 13) because GINC specifies a pre-trained distribution that is different from the prompt distribution. In GINC, there are no explicit input-output pairs; instead, the entire trajectory performs one task. Therefore, we encode the task vector into the inserted token "-" in the trajectory.

To evaluate in-context learning performance, we follow the setup described in Xie et al. (2021) and sample prompts from the five concepts used during training. An example of an in-context prompt with a context length of $k = 3$ and example length $s = 4$ is: a b c - d /

e f g - h / i j k - l. Similarly, an example zero-shot prompt with example length $s = 4$ would be: m n o - p. Here, the input $x$ represents a short document, such as a b c, and the task $f$ corresponds to an underlying hidden Markov model. The output $f(x)$ is the next sampled token in the document. Unlike the synthetic tasks described earlier, this function $f$ is stochastic.

To evaluate task vector prompting performance, we extract the hidden states at the i-th "-" token and inject them into the $l$-th layer activations of the zero-shot prompt at its "-" token. This setup aligns with the input format described in Hendel et al. (2023). Notably, because the first "-" token in the in-context prompt appears after an initial document of the HMM (e.g., the a b c document in the earlier example) and is then injected into the zero-shot prompt, additional context information from this HMM document may be included, which could lead to better task vector prompting performance compared to in-context learning performance. Through out the experiment, we set $N = 1$, i.e. the task vector is extracted from a single demonstrated prompt.

### B.2.2 The RegBench Dataset: the Probabilistic Finite Automata.

The RegBench dataset is introduced in Akyürek et al. (2024) to study the behavior of language pretraining and the emergence of n-gram head. We follow and revise the setup of Akyürek et al. (2024): we sample 100 deterministic finite automata (DFAs) for training. At each training iteration, we randomly generate 5,000 training sequences from these DFAs. Each DFA has a fixed initial state, and each state transits to another state given an input token. For each DFA, the total vocabulary size (i.e., the number of unique tokens) is set to 40, the maximum number of states is set to 20, and each state is allowed a maximum of 10 outgoing edges. To generate samples (sequences) from a DFA, we convert it to a probabilistic finite automaton (PFA) by assigning uniform transition probabilities across all outgoing edges, following Akyürek et al. (2024).

Each prompt contains 32 samples (sequences) generated by the sampled DFA and those example are with length $s$ separated by the special token "|". To help the formation of task vectors, we insert a dummy token ">" before the last token of the example. After inserting >, a prompt with context length $k = 3$ and example length $s = 4$ would be like "abc>d|efg>h|ijk>l".

We use the GPT-2 model with 8 layers, 2 heads, and an embedding dimension of 128 following Akyürek et al. (2024). For each training sequence, we sample an additional example from the same DFA and insert a dummy token > before the last token of the example. This serves as a zero-shot prompt. Similar to the GINC dataset, the input $x$ corresponds to the generated sequence of tokens from the PFA. The task $f$ is determined by the PFA's transition function, and the output $f(x)$ is the next token sampled by the DFA. Importantly, this transition function is stochastic as well. During training, we use the next token prediction loss on all tokens except the special tokens ">" and "|". During inference, we evaluate task vector prompting performance with $N = 1$.

## C   Supplements on the Trained-from-Scratch Model

### C.1   Effects of Input Formats

In this section, we examine different input formats for in-context learning prompts and their effects on the emergence of task encoding. The various prompt formats are detailed in Table 3. For clarity, we refer to the token positions where task vector presence is examined as the *task token locations*.

We focus on the case where the problem dimension is $d = 6$ and the model depth is $L = 3$, as it exhibits the most pronounced task encoding, as shown in Figure 2. In Figure 5, we evaluate various task vector extraction methods (detailed in Appendix C.2), with their respective task vector prompting performance shown as line with triangular markers. Notably, only the model with the input format (*xy) demonstrates task encoding that surpasses random

baselines. By default, we use the format (*xy) throughout the main paper unless specified otherwise.

### C.1.1 Attention Cluster Visualization

To understand the non-random performance observed in the task vector prompting mode, we analyze how the model encodes task information using attention maps (*left*) and a PCA of activations at each layer while varying the number of in-context examples (*right*), when input format is *xy in Figure 6 and when input format is x->y in Figure 7. In transformer models, attention mechanisms enable each input token to dynamically focus on specific parts of the input data by assigning importance to them. This process can be visualized using attention maps, which quantify the relationships between input tokens. These maps are computed using query and key vectors: the query represents the current focus of interest, while the key interact with the query to calculate importance scores (via their dot product), determining how relevant each input element is to the query.

Recall that we follow the experimental setup in Section 4.1 and conduct experiments on the linear regression task using a 3-layer transformer with a 63-shot context. On the left, the attention map of a specific head in 2nd layer shows that the activations at $x_i$ position primarily attend to the activations of the preceding $y_{i-1}$, where task information is stored. Furthermore, the activations at the $y_i$ location attend to both themselves and the preceding $y_{i-1}$, enabling the model to update task information online. On the right, we present a PCA of the activations for token $y_i$ ($i \in \{0, 2, 4, 6, 8\}$) across layers $L$, evaluated on three different linear functions: $w_1 = [1, 0, 0, 0, 0, 0]^T$, $w_2 = [0, 0, 1, 0, 0, 0]^T$, and $w_3 = [0, 0, 0, 0, 1, 0]^T$.

When the input format is *xy, at the input to the third layer, the activations for different tasks begin to form distinct clusters, though there is still considerable overlap and dispersion among the clusters. However, for x->y, across different layers, the attention maps do not exhibit meaningful task information encoded in the patterns. Consequently, we showcase only one example attention map to illustrate the absence of distinct task encoding in this setup. Additionally, the PCA of the activations reveals that the encoded information is nearly identical across tasks.

### C.1.2 Discrepancy with Pre-Trained LLM

In the pre-trained LLM (Hendel et al., 2023), the input format uses x->y. There is a clear difference between the pre-trained LLM and the trained-from-scratch transformer in how task information is encoded. In the pre-trained LLM, task information is primarily stored in the "maps-to" token (->). Conversely, in the trained-from-scratch transformer, task information is stored in the $y$ token (i.e., the "label" token), but only when x and y alternate closely in the input format.

We hypothesize that this discrepancy arises because pre-trained LLMs learn the semantic meaning of the "maps-to" symbol (->) from their extensive pre-training corpus, enabling them to use -> as an anchor for task summarization. Additionally, this delimiter -> is needed to separate the $x$ and $y$ entries. In contrast, for the trained-from-scratch transformer, the -> token functions more like a <pause> token (Goyal et al., 2023), providing additional computational resources rather than semantic significance. Moreover, the training procedure ensures that $x$ and $y$ tokens occupy fixed positions, removing the need for a delimiter to separate them.

Furthermore, Wang et al. (2023) show that label words themselves can serve as anchors for aggregating task information in context. This observation aligns with our finding that task information is encoded in the $y$ tokens. While this result is not explicitly framed within the task vector framework, it highlights the critical role of label tokens in task encoding.

### C.2 Various Task Vector Locating Methods

In this section, we analyze different methods for extracting task vectors. The main question we address is: *If task encoding is detectable by one task vector extraction method, will it also be detectable by another method?* Our focus is not on the quality of the extracted task vector but

Table 3: *Different prompt formats examined.* We analyze the input format based on the settings described in Garg et al. (2022) and Hendel et al. (2023). Here, $z$ represents a special token, $P_{\text{query}}$ denotes the test query format, and $P_k$ refers to the $k$-shot context. We examine the activations at $\Lambda_f$, referred to as the task token location.

| | $P_{\text{query}}$ | $k$-shot examples $P_k$ | Task Token Location $\Lambda_f$ |
|---|---|---|---|
| (*xy) (Garg et al., 2022) | $[z, x_{\text{test}}]$ | $[z, x_1, f(x_1), \cdots, x_k, f(x_k), x_{k+1}]$ | $P_k[0::2]$: $\{f(x_i)\}$ and $z$ |
| (x->y) Hendel et al. (2023) | $[x_{\text{test}}, z]$ | $[x_1, z, f(x_1), \cdots, x_k, z, f(x_k), z]$ | $P_k[1::3]$: $z$ |
| (*x->y) Hendel et al. (2023) | $[z, x_{\text{test}}, z]$ | $[z, x_1, z, f(x_1), \cdots, x_k, z, f(x_k), x_{k+1}, z]$ | $P_k[0::3]$: $\{f(x_i)\}$ and $z$ |

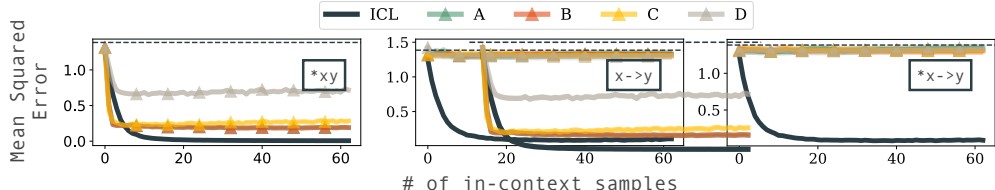

Figure 5: *In-context learning (ICL) and task vector prompting (TVP) performance across input formats and task vector extraction methods.* This figure compares task vector prompting performance across four task vector extraction methods (detailed in Appendix C.2) and three input formats: (*xy), (x->y), and (*x->y). Method (A) is the default task vector extraction method, and (*xy) is the default input format used throughout the main paper. As shown, for the (*xy) input format, noticeable task encoding is observed, with performance exceeding random predictions, regardless of the extraction method. In contrast, for the (x->y) and (*x->y) input formats, no task encoding can be reliably extracted by any method, resulting in task vector prompting performance that is nearly random.

rather on whether the method can detect task encoding that is non-random. Below, we present task vector extraction methods inspired by Hendel et al. (2023); Liu et al. (2023a); Li et al. (2024b), which are representative approaches in the task vector literature.

Following the definition in Section 3, we study the following task vector extractor methods:

**(A) Task Token's Output Embedding** Hendel et al. (2023) have observed that in pre-trained language models, the $l$-th layer output embedding of the $k$-th task token, denoted as $h_k^l$, encodes the task vector. We follow this setup and define the task vector extractor $g$ as follows: Let $\tau = h_k^l$, then

$$M(P_{\text{query}}; h_k^l) := \texttt{replace task token's l-th layer embedding with } h_k^l.$$

This notion of the task vector indicates the independence of the encoded task information from the demonstrated prompt.

**(B) Task Token's Output Embedding Difference** Liu et al. (2023a) pointed out that when inputting two sequences $x$ and $y$, the difference in the hidden space corresponds to the mapping from $x$ to $y$ in the pre-trained model. Though their setting is slightly different from the in-context learning format, we leverage this idea and propose the following way to extract the task embedding: Following the definition of $h_k^l$ from above, and let $\tilde{h}_k^l$ be the hidden state of the model when inputting the test prompt $P_{\text{query}}$, then $\tau = h_k^l - \tilde{h}_k^l = \Delta h_k^l$, indicating the difference between the $k$-th context embedding and an uninformative context, then

$$M(P_{\text{query}}; \Delta h_k^l) := \texttt{add } \Delta h_k^l \texttt{ to task token's l-th layer embedding.}$$

This notion of the task vector loosens the constraint of absolute independence of the task from the demonstrated prompt.

**(C) Principal Direction of Task Token's Output Embedding Difference** Liu et al. (2023a) also observed that taking the principal direction along the difference of hidden states

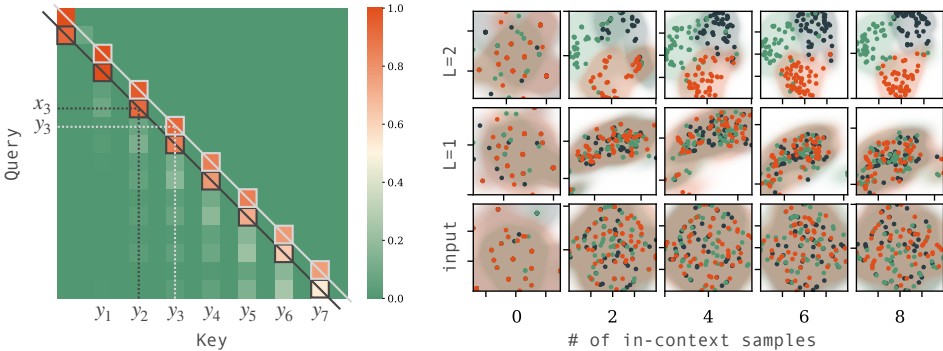

Figure 6: *Attention map and PCA visualization of activations across three different linear functions.* (**Left**) For the linear regression task described in Section 4.1, the attention map illustrates how each query (rows) attends to all available keys (columns), with each row summing to 1. The heatmap reveals that the activations at the $x_i$ positions predominantly attend to the activations of the preceding $y_{i-1}$, where task information is stored (highlighted by the black boxes). Additionally, the activations at the $y_i$ positions attend to both themselves and the preceding $y_{i-1}$, enabling the online updating of task information (highlighted by the white boxes). (**Right**) PCA visualizations of token $y_i$ activations ($i \in \{0, 2, 4, 6, 8\}$) across layers $L$ reveal that task-specific clusters (three colors correspond to three different tasks) begin to emerge at the output of the 2nd layer, indicating that the model progressively encodes task information as depth increases.

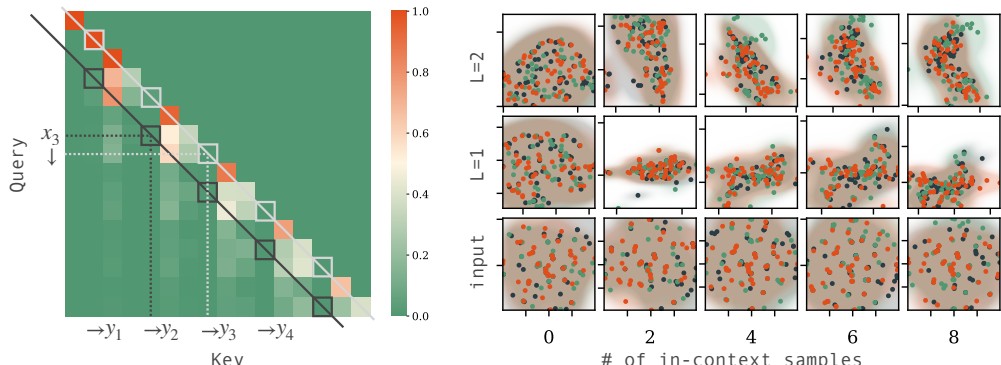

Figure 7: *Attention map and PCA visualization of activations for three linear functions with input format* (x->y). Following the same setup as in Figure 6, we plot the attention map and PCA visualization of the -> token's activations. (*Left*) The attention map at the 2nd layer highlights interactions between task tokens (white boxes) and the attention from $x_i$ tokens to task tokens (black boxes). Unlike the format (*xy), this format does not exhibit a meaningful attention pattern where $x_i$ tokens attend to previous task tokens, or task tokens attend to prior task tokens. (*Right*) PCA visualizations show that the activations of the -> token encode nearly identical information across different context lengths and different tasks, suggesting the absence of distinct task encoding dynamics in this input format.

aligns more closely with the task. Motivated by their observation and setting, we propose the following task vector extractor: Following the definition of $\Delta h_k^l$ from above, we set $\tau = \text{PCA}(\Delta h_k^l)$, then

$$M(P_{\text{query}}; \text{PCA}(\Delta h_k^l)) := \text{add PCA}(\Delta h_k^l) \text{ to task token's l-th layer embedding.}$$

We also re-scale the updated hidden states to preserve the model's original capabilities. This approach differs slightly from the original setup in Liu et al. (2023a), where PCA is applied to the concatenated hidden states across all layers. Their method is specifically designed for alignment tasks in natural language settings, where the context $x_i$ and $y_i$ are input separately. In our case, as the model is trained in an in-context learning (ICL) format, we adhere to the input prompt structure where $x_i$ and $y_i$ are provided alternately in one prompt. We also attempted to follow their setup by applying PCA to the concatenated

hidden states across all layers; however, this approach resulted in near-random task vector prompting performance.

**(D) Linear Combination of Task Token's Output Embedding to the Query Output Embedding** Li et al. (2024b) propose to learn the coefficient of the linear combination between the task token's output embedding to the query output embedding. Motivated by this, we let $\tau = \{h_k^l\}_{l \in [L]}$, then

$$M(P_{\text{query}}; \{h_k^l\}_{l \in [L]}) := \texttt{replace task token's l-th layer embedding with}$$
$$\alpha_l^h h_k^l + \beta_l^h h_{\text{test}}^l \quad \forall l \in [L].$$

where $h_{\text{test}}^l$ is the output embedding at $l$-th layer for the test query, and $\alpha_l^h$, and $\beta_l^h$ are learnable parameters. To learn these coefficients, we use a constant learning rate of 0.01, the AdamW optimizer, and train for a total of 100 epochs. Following the initialization setup in Li et al. (2024b), the coefficients are initialized as $\alpha_l^h = 0.1$ and $\beta_l^h = 1$.

### C.2.1 Task Vector Extractor Performance

We evaluate the performance of various task vector extraction methods across different input formats, as described in C.1, and present the results in Figure 5. When using the *xy input format, the task vector extraction methods demonstrate noticeable task encoding. However, for the other input formats, no informative task encoding is detected, regardless of the extraction method used.

It is worth noting that the learnable task vector extraction method (D) performs slightly worse than the other three methods, possibly due to its initialization. We did not focus on further improving this method, as the primary goal of this section is to demonstrate the detectability of task encoding rather than to optimize its quality.

### C.3 Formal Description of Layer Localization for Task Vector

For each task $f$, we generate $N$ demonstrated prompts $\{P_k^{D,j}\}_{j=1}^N$, where the same function $f$ is consistently used across all $N$ prompts, and the input to each prompt $P_k^{D,j}$ is uniformly sampled. Let $P_{\text{query}}^j$, with test input $x_{\text{test}}^j$, represents the corresponding test queries for the $j$-th demonstrated prompt. For a task vector extracted from a $k$-shot prompt, the optimal layer index $l^\star$ is computed as:

$$l^\star = \arg\min_l \sum_{j=1}^N \ell\left(f(x_{\text{test}}^j), M\left(P_{\text{query}}^j; g(M(P_k^{D,j}), l)\right)\right),$$

where $g(M(P_k^{D,j}), l)$ denotes the task vector extracted from the $l$-th layer when processing the demonstrated prompt $P_k^{D,j}$, and $M(P_{\text{query}}^j; \tau)$ represents the model's prediction for the query prompt $P_{\text{query}}^j$, where the task vector $\tau$ extracted from the $l$-th layer is copied to the corresponding position in the model's representation when processing $P_{\text{query}}^j$. A lower loss $\ell$ indicates that the corresponding layer provides a more effective encoding of task-specific information.

### C.3.1 Task Vector Layer Localization in Linear Regression Tasks

When extracting the task vector, the extractor must determine which layer the task vector resides in, based on the $N$ demonstrated prompts. For linear regression task with the experimental setup described in Section 4.1 and Appendix B.1, we measure model's performance in the TVP mode when the extracted task vector is placed at each layer and average the loss across varying context lengths. Figure 9 (left) demonstrate that among the 3 layers, the 2nd layer is predominantly selected as the location where the task vector resides.

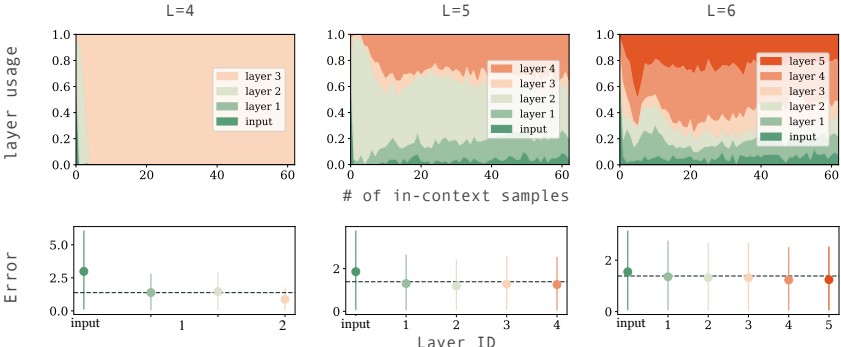

Figure 8: *Layer selection distribution (upper row) and averaged task vector prompting (TVP) performance (lower row) for task vector emergence in Linear Regression with problem dimension $d = 6$ and model depths $L = 4$ (left), $L = 5$ (middle), and $L = 6$ (right).* **Upper row:** The frequency of each layer being selected as the task vector location across varying numbers of in-context examples is shown. As model depth increases, task information becomes less concentrated in any single layer, resulting in a more distributed selection of task vector locations. **Lower row:** The averaged TVP performance for each layer, measured across various context lengths, is presented. The dashed line indicates random-guess performance, representing a scenario where no task information is inferred. These results demonstrate that for deeper models (especially $L = 5, 6$), no single layer consistently encodes significant task information, and task vector performance approaches random, highlighting the distributed nature of task encoding in deeper architectures.

This observation contrasts with the phenomenon reported in pre-trained LLMs by Hendel et al. (2023), where task vectors for in-context learning tasks primarily emerge in the early layers out of a total of around 20 layers. In our experiments, however, we find that task vectors consistently emerge in the penultimate layer for shallow models (e.g., 3 layers). As the model depth increases (e.g., 4 or more layers), task vectors no longer emerge distinctly at any single layer, as shown in Figure 2. Instead, task information becomes distributed across multiple layers, reducing the distinctiveness of any specific layer as the task vector's location. As shown in Figure 8, for deeper models, particularly with $L = 5$ and $L = 6$, no single layer consistently encodes task information, and the task vector prompting performance appears random. From this analysis, we conclude that in the linear regression task, and under the experimental setup we examined, task vectors either emerge in the penultimate layer when the model is shallow or fail to emerge entirely when the model is deep.

One factor to consider is the nature of the task at hand. For the linear function class, the "task" appears to be identifying the parameter $w$, after which the "task execution" phase simply requires the model to compute $w^T x$ to generate predictions. To explore this further, we examine the sinusoidal regression task using the same GPT-2 model configured with an embedding size of 64, 4 attention heads, and 6 layers. Additionally, we analyze the discrete token offset task with a larger embedding size of 256.

As shown in Figure 9 (middle), for this 6-layer transformer model, the 3rd layer is primarily selected as the task vector location. This finding aligns with observations in pre-trained LLMs and provides the insight that the task vector's emergence layer in trained-from-scratch models depends on the characteristics of the task being performed. Additionally, for the discrete token offset task, the 5th layer is most frequently selected as $l^\star$, with other layers also being selected occasionally at lower probabilities.

## C.4   Effects of Model Capacity

In this section, we examine the model with same depth, but different embedding size, to investigate the effect of model capacity. Specifically, for the three tasks mentioned in Section 4.1, we study the trained-from-scratch model's performance on model depth 4, 6 or 8, with embedding size 64 (with notation "S") or 256 (with notation "L"), and present the result in Figure 10.

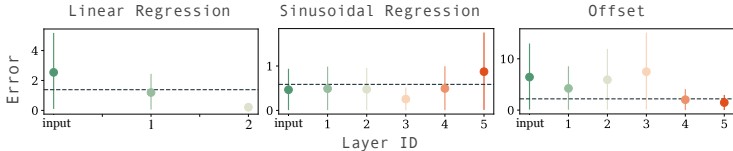

Figure 9: *Task vector prompting performance for task vector emergence in: Linear Regression (left), Sinusoidal Regression (middle), and Discrete Token Offset (right).* We present the averaged task vector prompting (TVP) performance for each layer, measured across various context lengths. The dashed line represents random-guess performance, indicating no task information is inferred.

As illustrated in the figure, increasing the embedding size allows the model to discover shortcut solutions for regression tasks. Specifically, the model approximates the least squares solution in a single forward pass, eliminating the need to store intermediate task encodings. Conversely, for the discrete token offset task, the model operates on tokens represented in a learned embedding space. To solve $f(x) = a \times x + b$, the model must map one token to another using the task-specific parameters $a$ and $b$. We hypothesize that the nature of token embeddings makes it less straightforward for the model to learn a shortcut solution for this task. As a result, the model is compelled to store the task information explicitly. Consequently, increasing the model capacity provides more space for storing and organizing task encodings, leading to clearer task representations as capacity grows.

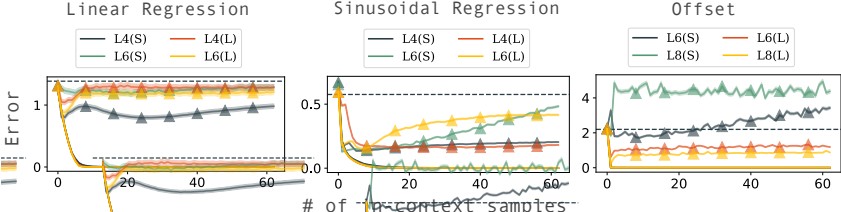

Figure 10: *Effects of model embedding dimension on task vector prompting performance.* This figure shows the task vector prompting performance of models with different embedding sizes (64 for notation "S" and 256 for notation "L"). For the regression tasks (Linear and Sinusoidal), larger embedding sizes make task encoding less prominent, likely because the increased capacity allows the model to find "shortcut" solutions, such as approximating the least-squares solution directly, without the need to store intermediate task information. In contrast, for the discrete token offset task, a larger embedding size enables the model to store task information in a more separable and structured manner.

## C.5  Effects of Sparse Attention

Sparse attention (Lou et al., 2024) has been proposed to improve transformer efficiency with minimal performance degradation. In in-context learning tasks (detailed in Section 4.1), a model could potentially solve these tasks using sparse attention: For each query, the model only needs to attend to itself and the previously stored task information. Specifically, each $x_i$ token needs to attend to two tokens ($x_i$ and $y_{i-1}$), while each $y_i$ token needs to summarize the task information by attending to three tokens ($y_i$, $y_{i-1}$, and $x_i$).

In this section, we investigate the impact of explicitly enforcing sparse attention constraints on in-context learning and task vector prompting performance. Using the setup described in Section 4.1, we apply a sliding window strategy (Qiu et al., 2019; Beltagy et al., 2020) combined with causal attention, with window sizes $s$ of 3, 5, and 7. We evaluate the transformer with various depth on the linear regression task with $d = 6$ (where task vectors emerge) and $d = 9$ (where task vectors are nearly absent), and present the results in Figure 11.

As shown in the figure, models trained with sparse attention demonstrate better task encoding compared to those with full-window causal attention. For instance, when $d = 6$ and $L = 5$, task encoding in the full-window model fades after the first few in-context

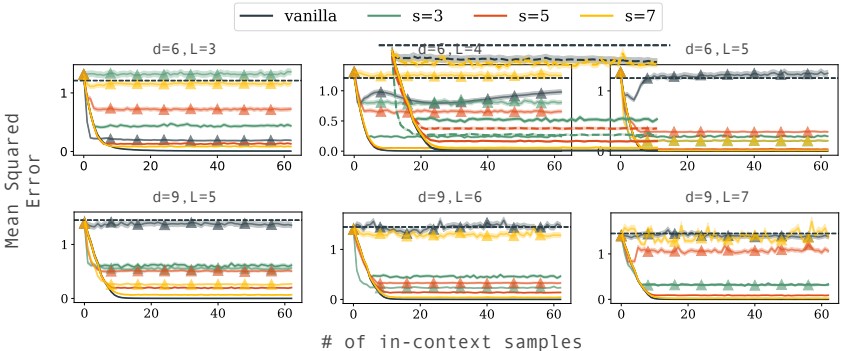

Figure 11: *Task vector prompting performance for models trained with sliding window sparse attention. We compare models trained with full-window causal attention (vanilla) and sparse attention using sliding window sizes of 3, 5, and 7. Solid lines represent the in-context learning performance, while lines with extra triangular marker indicate the task vector prompting performance. As shown, sparse attention enhances task vector emergence but leads to a slight degradation in in-context learning performance.*

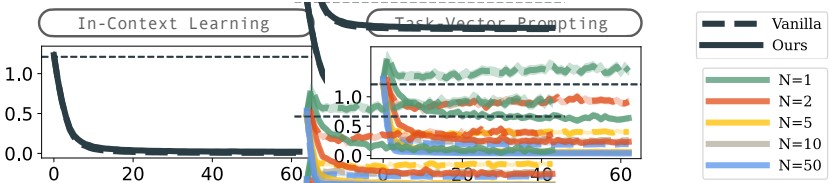

Figure 12: *Impact of the Number of Demonstrated Prompts (N) on Task Vector Prompting Performance. Increasing N improves task vector prompting performance, as larger N values yield more concentrated task vectors. Across all N settings, model trained with TVP-loss ("Ours") consistently outperforms the vanilla model, demonstrating better task vector encoding. The dashed line represents random-guess performance.*

examples. In contrast, models trained with sliding window attention (window sizes 3, 5, and 7) maintain stable task encoding across varying context lengths.

However, while the sliding window improves task vector emergence, it slightly degrades in-context learning performance because each query has access to fewer tokens during training. Moreover, task encoding stability is not always guaranteed: larger window sizes improve in-context learning performance but degrade task vector prompting performance.

### C.6 Effects of Number of Demonstrated Prompts $N$

Throughout the paper, we use $N = 50$ by default when evaluating task vector prompting performance. Intuitively, larger $N$ values yield more concentrated and purified task vectors. To illustrate the effect of $N$, we evaluate task vector prompting performance for $N = 1, 2, 5, 10, 50$, following the setup in Section 4.1, and present the results in Figure 12. In this experiment, the transformer is trained on a linear regression task with a problem dimension of $d = 6$ and transformer depth $L = 3$. For the model trained with TVP-loss, the task vector is explicitly formed at the 2nd layer.

As shown in the figure, when $N = 1$, the vanilla model yields random task vector prompting performance. With increasing $N$, task vector performance improves steadily. Notably, the model trained with TVP-loss (denoted as "Ours" in the figure) demonstrates better task vector encoding compared to the vanilla model across all $N$ values. Because the linear functions in regression tasks have weights that exist in a continuous space and can be very close to each other, separating tasks with only a single demonstration is challenging. As a result, it requires $N = 5$ to achieve task vector prompting performance comparable to its in-context learning performance. For the vanilla model, however, even with $N = 50$, task vector prompting performance remains slightly worse than in-context learning performance.

## D    Supplements on Training with TVP-Loss

### D.1    Demonstrating Our Method in Meta-ICL Training / Fine-Tuning

We illustrate the training algorithm in Figure 13, where the red arrow indicates the flow of the gradient.

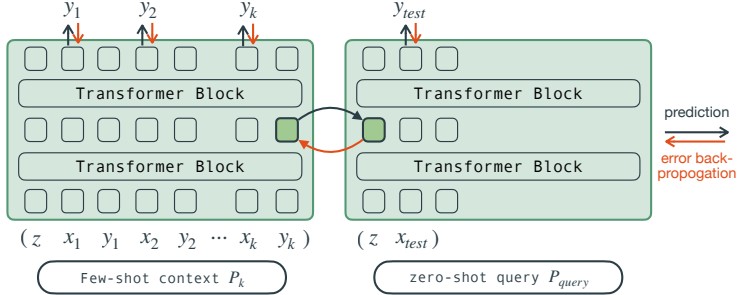

Figure 13: *Demonstration of our training algorithm.* In vanilla Meta-ICL training, the model is updated using the ICL-loss signal from the *few-shot context*. To encourage the formation of task vectors, we also explicitly include the TVP-loss from the *zero-shot query*. This means the model is asked to predict $y_{\text{test}}$ when only $x_{\text{test}}$ and the injected hidden states are given. In the given illustrated example, there are in total 2 layer in the transformer model, and we set $l = 1$ to encourage the formation of the task vector at the first transformer block's output.

### D.2    Full Results for Section 5.2

In Section 5.2, we present the in-context learning and task vector prompting performance at the 63rd context position for models trained with and without the TVP-loss, along with the in-context learning performance on out-of-distribution prompts. Figure 14 extends these results by showing the full performance across varying context lengths. This analysis complements the main results, demonstrating that the observations at the 63rd position are not unique but instead hold consistently across different context lengths.

### D.3    Performance on Additional OOD Tasks

For the experiment result presented in Section 5.2, we evaluate the model's performance on additional out-of-distribution (OOD) prompts for the regression tasks, specifically: (a) logistic regression, (b) in-context learning with an outlier context presented with a probability of 0.1, where the outlier is defined as $x = 1$ and $y = 1$. (c) in-context learning with noisy labels (noise level 0.3), (d) scaled weights ($\times 3$).

For the linear regression training task, models with task vectors formed at the 7th layer (i.e., the last layer) perform equal to or slightly better than the vanilla-trained model, except when the weights and inputs are scaled. This suggests that models trained with TVP-loss exhibit reduced robustness when the input range is significantly altered. For the sinusoidal regression task, models with task vectors formed at the 3rd layer exhibit equal or improved OOD performance.

### D.4    In-Context Multi-Function Learning

To further explore compositional tasks, we investigate the model's ability to generalize to unseen functions by training on multiple compositional functions. Specifically, we follow the linear regression task's sampling strategy to sample $w$ and train on the following function classes:

1. *Square-Root Regression*: $f(x_i) = \sqrt{\max(0.5 \cdot w^\top x_i + h, 0)} + b$, where $h, b \sim \mathcal{N}(0, 1)$.

2. *Sinusoidal Regression*: $f(x_i) = \sin(0.5 \cdot w^\top x_i + b)$, where $b \sim \mathcal{N}(0, 1)$.

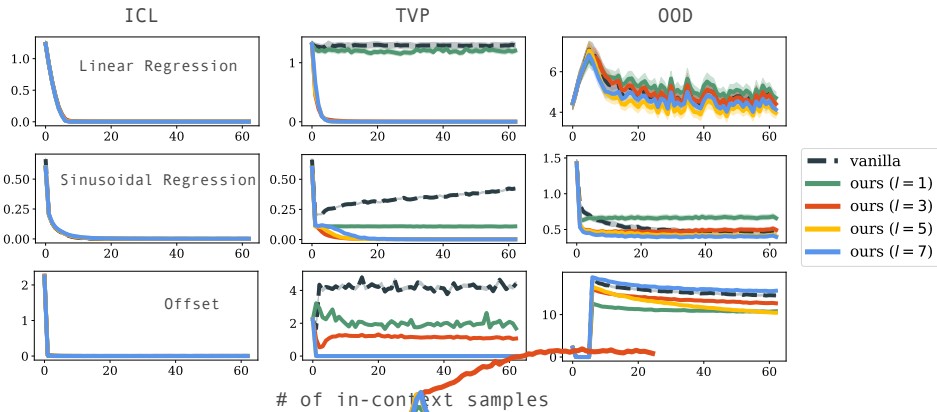

Figure 14: *Model Performance with Vanilla and TVP-Loss Training.* We compare the in-context learning (ICL), task vector prompting (TVP), and out-of-distribution (OOD) performance of models trained with vanilla training (black) and those trained with our TVP-loss, where task vectors are formed at different layers ($l = 1, 3, 5, 7$). Full results are shown across varying numbers of in-context examples for three tasks: linear regression (top row), sinusoidal regression (middle row), and discrete token offset task (bottom row).

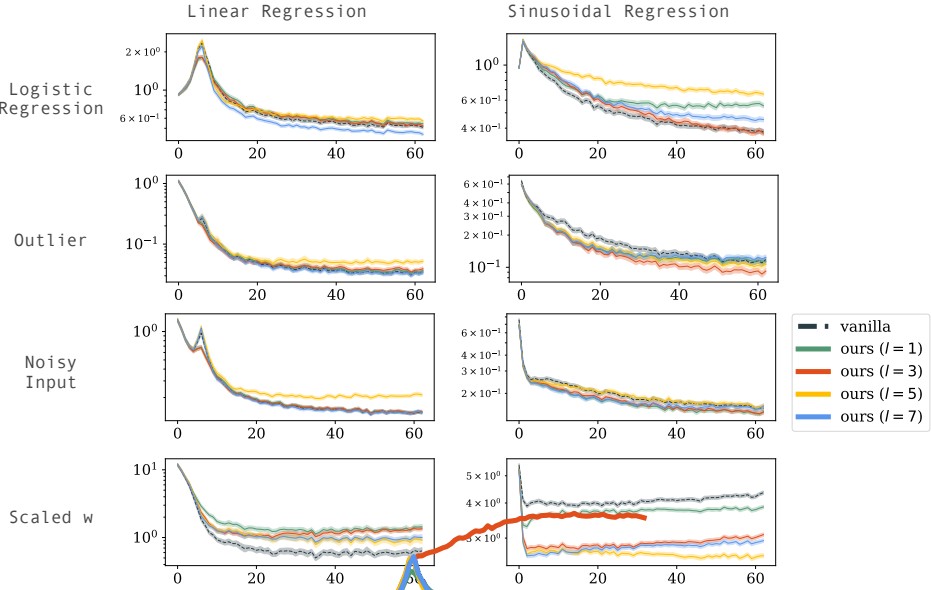

Figure 15: *In-Context Learning Performance on OOD Prompts for Models Trained with Regression Tasks.* We compare the in-context learning performance of the vanilla-trained model (labeled as *vanilla*) and models trained with auxiliary loss to form task vectors at different layers $l$ (labeled as *ours (l)*). We evaluate the models on OOD prompts, including: (a) logistic regression, (b) outlier in context, (c) noisy input labels with a noise level of 0.3, (d) scaled weights ($\times 3$).

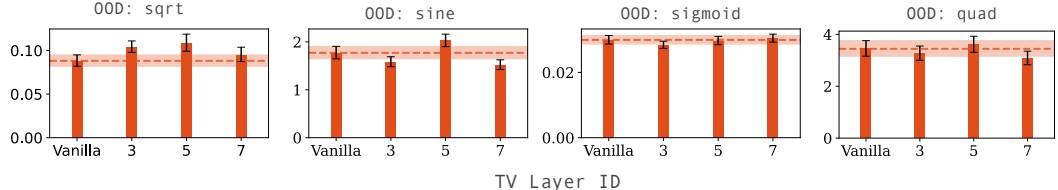

Figure 16: *Out-of-Distribution (OOD) Performance with Vanilla and TVP-Loss Training.* This figure compares the OOD prompt performance of models trained with vanilla training and TVP-loss across layers where task vectors are formed. Dashed lines represent the performance of vanilla-trained models for reference. The horizontal axis indicates the layer in which task vectors are formed. TVP-loss improves OOD generalization, particularly for the sine and quadratic regression tasks, while achieving comparable performance to vanilla training for the sigmoid task, and slightly worse performance on square root regression task..

3. *Sigmoid*: $f(\boldsymbol{x}_i) = \frac{1}{1+\exp(-0.5\cdot\boldsymbol{w}^\top\boldsymbol{x}_i+b)}$, where $b \sim \mathcal{N}(0,1)$.

4. *Quadratic Regression*: $f(\boldsymbol{x}_i) = 0.5(\boldsymbol{w}^\top\boldsymbol{x}_i - h)^2 + b$, where $h, b \sim \mathcal{N}(0,1)$.

During training, we designate one function $i \in \{1, 2, 3, 4\}$ as the out-of-distribution (OOD) function and train the transformer using in-context prompts sampled from the remaining three function classes, each sampled uniformly. We then evaluate the model's compositional generalization on the OOD function.

We train an 8-layer transformer with 4 attention heads and an embedding dimension of 256, keeping all other training hyperparameters consistent with Section 4.1. To examine the effect of task vector location, we test placing the task vector at the 3rd, 5th, or 7th layer. The results, shown in Figure 16, represent the averaged in-context learning performance across context lengths. These results demonstrate that TVP-loss enhances OOD generalization, particularly for the sine and quadratic regression tasks, achieves comparable performance to vanilla training for the sigmoid task, and shows slightly reduced performance for the square root regression task.

## D.5  Indentification of task vector in GINC dataset

In GINC, there are no explicit input-output pairs; instead, the entire trajectory performs one task (hidden markov model). Therefore, we encode the task vector into the inserted token "−" in the trajectory. We provide an illustration of our training algorithm in Figure. 17.

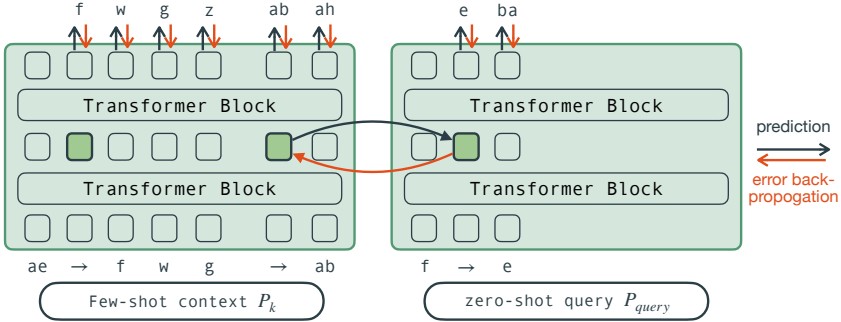

Figure 17: *Demonstration of our training algorithm in GINC dataset.* In contrast to the Meta-ICL format described in Figure 13, in GINC dataset, there are no explicit input-output pairs. Instead, the entire trajectory performs one single task defined by a hidden markov model. Therefore, we randomly insert several →" tokens into the trajectory in the few-shot mode, while inserting one →" token into the trajectory in the zero-shot mode. The task vector at the $l$-th layer is then injected into the embedding in the zero-shot mode.

### D.6 Fully Converged Performance on the RegBench Dataset

As noted by Reviewer GXZE, our results on the RegBench dataset (Figure 4) were based on undertrained models. To address this concern, we double the training duration and report updated performance in Figure 18 and Figure 19.

Figure 18 shows the next-token prediction accuracy over the course of training. Both the baseline and our TVP-loss model converge by epoch 40, indicating comparable performance at converging. However, as shown in Figure 19, our method continues to demonstrate better ICL and TVP performance, especially at prompt length $s = 4$. These results suggest that TVP-loss provides a structural benefit by encouraging more robust and transferable task representations, leading to stronger ICL generalization.

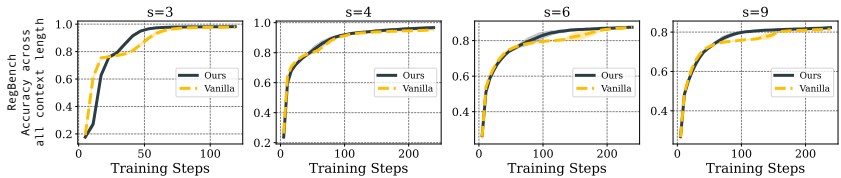

Figure 18: *Training accuracy on the RegBench dataset over 40 epochs.* Both the baseline and TVP-loss models converge in next-token prediction.

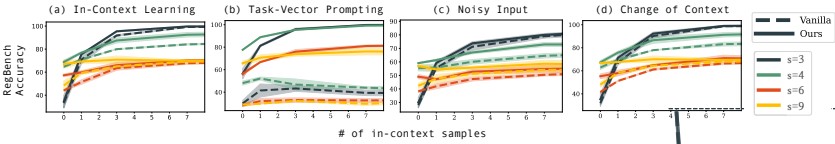

Figure 19: *In-context learning (ICL) accuracy on RegBench at convergence.* The TVP-loss model consistently outperforms the baseline across different prompt lengths, particularly at $s = 4$.

