# OpenReview forum: "Task Vectors in In-Context Learning: Emergence, Formation, and Benefits"
_colmweb.org/COLM/2025/Conference — COLM 2025_

### Official Review · Reviewer_g1s8 · 2025-05-11

**Rating:** 6
**Confidence:** 3
**Ethics Flag:** 1

**Summary:**

This paper studies the emergence of task vectors during LM training, and then proposes a loss to encourage the formation of strong task vectors. Experiments show that the proposed loss further improves models ICL performance and the generalization ability on OOD data.

The paper is overall well-written, and the visualizations are clear. The proposed loss is well-motivated with validation on experiments.

**Questions To Authors:**

1. In Fig 3, which base model does TVP evaluation use at the inference time? The model $M _\theta$ trained together with in-context loss or the untrained base model $M$?

2. It could be helpful to provide more insights on why adding the TVP loss can improve the ICL/MetaICL performance.

**Reasons To Accept:**

1. The problem of studying task vectors and using them to enhance prompting is important and can benefit LM tuning efficiency.

2. The paper is well-written, with experimental settings and results clearly described. The experiments are thorough, conducted on both synthetic and real datasets.

3. The proposed loss is well-motivated, and the experiments further validate its efficacy.

**Reasons To Reject:**

1. The experiments on natural language tasks are only conducted under the MetaICL setting. What are the reasons behind that and how does the proposed loss perform in the general ICL/TVP setting?

---

> ### Author Response · Authors · 2025-06-03
>
> We thank the reviewer for their thoughtful and constructive feedback. We are glad that the importance of studying task vectors, the clarity of our writing and visualizations, and the effectiveness of the proposed loss are recognized and appreciated. Below we address the reviewer’s concerns in detail.
>
> > Q1. The experiments on natural language tasks are only conducted under the MetaICL setting. What are the reasons behind that and how does the proposed loss perform in the general ICL/TVP setting?
>
> We appreciate the question and the opportunity to clarify. The MetaICL setting refers to fine-tuning a pretrained language model on a diverse set of in-context learning (ICL) tasks, as introduced in [1], and then evaluating the model on held-out ICL tasks. In our experiments, both our method and the vanilla MetaICL baseline follow this process. The only difference is that our method incorporates the TVP-loss during fine-tuning. Evaluation is performed in ICL mode for both methods.
>
> Since these held-out ICL tasks are not seen during training, and we do not explicitly supervise task vector formation for them, it is not expected that the model would perform strongly in TVP mode. However, we did evaluate TVP performance in the hr→lr setup. Specifically, we evaluated the TVP loss across 10 prompts to select the optimal layer index $l$, and then used the embedding from layer $l$ to perform task vector prompting for the remaining prompts within the task.
>
> While the vanilla MetaICL model achieved a TVP performance of $31.81 \pm 0.13$, our method yielded $34.15 \pm 0.14$, suggesting that TVP-loss slightly improves task localization ability—even on tasks that were not explicitly supervised during training.
>
>     [1] Min, Sewon, et al. "Metaicl: Learning to learn in context." arXiv preprint arXiv:2110.15943 (2021).
>
> > Q2. In Fig 3, which base model does TVP evaluation use at the inference time?
>
> In Figure 3, TVP evaluation is conducted using the models trained with the standard ICL loss (i.e., the $M_{\theta}$ model) for the vanilla baseline, and with both ICL loss and TVP-loss for our method.
>
> > It could be helpful to provide more insights on why adding the TVP loss can improve the ICL/MetaICL performance.
>
> We appreciate this important question. Our insight is that TVP-loss introduces an architectural bias that guides the model to develop localized and consistent task-specific representations at particular layers. This design aligns with our empirical finding that such structured representations naturally arise in moderately sized models. In this sense, TVP-loss amplifies a tendency that the model already exhibits when its capacity is appropriately matched to the task.

---

> > ### Comment · Reviewer_g1s8 · 2025-06-09
> >
> > Thank the authors for the detailed explanation. I maintain my score.

---

### Official Review · Reviewer_GxZE · 2025-05-12

**Rating:** 6
**Confidence:** 4
**Ethics Flag:** 1

**Summary:**

The paper looks into the formation of task vectors in small transformer models trained on synthetic datasets, and studies this over a set of hyperparameters and input structures. It also proposes a loss to aid in the formation of task vectors in such models.

**Questions To Authors:**

- Since adding the tvp loss supposedly increases robustness, do you expect the inverse to be true? Would techniques to improve robustness affect the tvp performance?
- What is the performance of the Natural Language Task if you choose and set a single layer as the other experiments?
- Why is vanilla transformer performance low in sections 5 and 6?
- Is the performance on OOD prompts in 5.2 checked in ICL or TVP mode?

**Reasons To Accept:**

- Current work focuses on looking at task vectors in pre-trained LLMs, so there is novelty in studying them in smaller transformers.
- There are clean experiments conducted to show the emergence of task vectors over the course of training transformers. There are sufficient ablations to show the effect of multiple model and training hyperparameters.

**Reasons To Reject:**

- It is somewhat obvious already that adding a loss that is identical to the metric being tested (TVP performance) will improve on that metric. In this regard the new loss seems to not be an insightful discovery. It would improve the paper to show other benefits of adding this loss and improving tvp performance. The OOD performance is an attempt at this but as highlighted later, it is not as convincing.
- It is quite unclear to me why the vanilla models are significantly underperforming in the experimental sections 5 and 6. As noted on line 248, in figure 3 the tvp performance of the vanilla model is near random. This is a surprising result because regularly trained transformers do exhibit evidence of task vectors, even in this paper itself in Figure 1. This extreme result is cause for concern. Similarly in section 6 it is not clear why the vanilla model should perform worse on ICL.
- The benefits of the new loss on OOD are unconvincing. In section 5.2 and figure 3, all (but a few) of the results are near random performance. Even the ones that do perform better than vanilla are not significantly better. In light of the close-to-random performance this result seems not indicative of an improved model.

  In Section 6 and figure 4, again the claimed result only seems to be true for a few settings, s=6 and s=9. As noted before, for these specific results the vanilla model seems to be undertrained even for ICL performance, and the OOD results simply mirror this lack of performance.

  In summary, both experiments are not convincing enough to claim an improvement in OOD generalization as a result of the TVP loss.
- What is the performance of the Natural Language Task without implicitly selecting the best layer to minimize TVP loss? How is this layer chosen at inference time?
The current setting serves as an unfair comparison to the baseline, as there is implicit training on validation performance in this method. The extra overhead of calling the model 8 times to check performance on the test query is significant.

---

> ### Author Response · Authors · 2025-06-03
> **Response (1/2)**
>
> We thank the reviewer for their thoughtful and constructive feedback. Below we address the reviewer’s concerns in detail.
>
> > Q1. Adding a loss that is identical to the metric being tested (TVP performance) will improve on that metric is obvious.
>
> Thank you for the comment. While optimizing a loss aligned with the evaluation metric can lead to expected gains, our key contribution is showing that adding TVP-loss during ICL training (1) improves TVP performance, (2) does not **degrade ICL performance**, and (3) **can improve OOD generalization**. To our knowledge, this observed benefits are non-trivial and require empirical validation.
>
> > Q2. **TVP** performance of Vanilla model in Section 5 are nearly random.
>
> Thank you for this insightful observation. As discussed in Figure 2, the emergence of task vectors is closely tied to model capacity. Specifically, task vectors tend to emerge only when the model is not overparameterized relative to the training dataset.
>
> In Figure 1, the model has 3 layers, which is sufficient to learn the task but not deep enough to bypass the need for explicit task representation. In contrast, Figure 3 uses an 8-layer model for the linear regression task. This deeper model has enough capacity to solve the task directly in a single forward pass, without needing to encode task information in intermediate representations. As a result, the TVP performance of the vanilla 8-layer transformer is near random.
>
> > Q3. Is the performance on OOD prompts in 5.2 checked in ICL or TVP mode?
>
> The performance on OOD prompts in Section 5.2 and throughout this paper is evaluated in ICL mode.
>
> > Q4. In Section 6, vanilla model in RegBench seems to be undertrained.
>
> We appreciate this observation. We extended training for both the vanilla model and our method, and updated the ICL results in Figure 6 [here](https://anonymous.4open.science/r/task_vector_prompting-85F8/rebuttal_figures/regbench_icl_performance_converged.pdf). To confirm convergence, we provide training accuracy curves [here](https://anonymous.4open.science/r/task_vector_prompting-85F8/rebuttal_figures/regbench_train_acc.pdf), showing the mean DFA accuracy across context lengths converges.
>
> In the updated figure, we still observe a gap in ICL and OOD performance, particularly when s=4. We believe the ICL and OOD improvement on RegBench reflects that TVP-loss helps structure the learning of these ICL tasks.
>
> We also want to clarify that TVP-loss is not designed to explicitly optimize for OOD performance. Rather, the observed improvement is a byproduct of encouraging structured task representations through task vector formation.
>
> > Q5. In the natural language task, how is the model evaluated during inference?
>
> During inference, we follow the ICL-style evaluation protocol from the original MetaICL paper [1]. For each test query, the model evaluates the conditional log-likelihood of each candidate answer given the prompt, and selects the answer with the lowest loss.
>
> > Q6. How is this layer chosen at inference time?
>
> Since the model is evaluated purely in ICL mode on various held-out datasets, no task vector layer is selected or applied during inference. We also discuss the potential of reporting the TVP performance of the OOD prompt in response to Reviewer g1s8 Q1.
>
> > Q7. There is implicit training on validation performance in this method.
>
> We clarify that the validation set is strictly held out from the training set. The decision to use adaptive layer selection was made based on its performance on the hr→lr task as opposed to select a single layer (as shown in Q8). For the other two tasks, we applied the same hyperparameter setup without further tuning.
>
> We want to note that our goal is to study if there is any benefit using TVP-loss in the ICL performance of the large-scale natural language models.
>
> > Q8. What is the performance of the Natural Language Task if you choose and set a single layer as the other experiments?
>
> Thank you for this insightful question. In the hr->lr task, we conducted additional experiments where the task vector location was pre-assigned to specific layers: 12, 14, 16, 18, and 20. The results are as follows:
>
> |vanilla|ours(adaptive selection)|ours(L=12)|ours(L=14)|ours(L=16)|ours(L=18)|
> |-|-|-|-|-|-|
> |44.66|45.38|44.24|41.53|42.48|32.93|
>
> In the MetaICL setup, the fine-tuning dataset contains a variety of tasks. Fixing the task vector at the 12th layer gives similar performance to vanilla, but performance drops consistently at deeper layers. We hypothesize that this is because different tasks encode best at different layers, and fixing the location limits the model’s ability to represent all tasks effectively. Adaptive selection during training provides the flexibility needed to capture task-specific information more robustly.

---

> > ### Author Response · Authors · 2025-06-03
> > **Response 2/2**
> >
> > > Q9. The extra overhead of calling the model 8 times to check performance on the test query is significant.
> >
> > We acknowledge this overhead. During training, candidate layer selection is performed in evaluation mode (i.e., without gradient computation), followed by a single gradient-tracked forward pass to compute the TVP-loss. Under mixed-precision training, this results in roughly a 3× training-time overhead.
> >
> > We note that no effort was made to optimize efficiency. We leave such improvements to future work, as our current focus is to characterize the emergence of task vectors and demonstrate their benefits.
> >
> > > Q10. Since adding the tvp loss supposedly increases robustness, do you expect the inverse to be true? Would techniques to improve robustness affect the tvp performance?
> >
> > We appreciate this thoughtful question. While we have not explicitly studied the reverse direction—i.e., whether robustness-enhancing techniques improve TVP performance—our intuition is that there could be some connection, but it is not guaranteed.
> >
> > TVP-loss works by explicitly encouraging the formation and localization of task-specific representations, which in turn improves robustness to prompt or task variation (e.g., OOD generalization). Some robustness techniques, such as data augmentation, may implicitly increase task diversity or complexity, potentially pushing the model to form more localized task representations—thus possibly enhancing TVP performance. On the other hand, techniques like dropout may encourage the use of more distributed representations, which could interfere with the emergence of strong task vectors and potentially reduce TVP performance.

---

> > ### Comment · Reviewer_GxZE · 2025-06-05
> > **Response to rebuttal**
> >
> > I am convinced by the authors' response and am increasing my score

---

> > > ### Author Response · Authors · 2025-06-06
> > >
> > > Hi Reviewer GxZE,
> > >
> > > Thank you for reviewing our response and for raising your score from 4 to 6. We truly appreciate your thoughtful review and support.

---

### Official Review · Reviewer_BR3X · 2025-05-12

**Rating:** 6
**Confidence:** 4
**Ethics Flag:** 1

**Summary:**

This paper discusses task vector formation in transformers​ for ​improving robustness and generalization in in-context learning (ICL)​. The paper (1) claims that task vectors emerge weakly and non-locally in pre-trained models, limiting their utility, and (2) proposes a ​task vector prompting loss (TVP-loss) to explicitly guide their formation during training. Experimental results on ​synthetic regression, formal language, and natural language tasks​ show that TVP-loss significantly enhances zero-shot performance compared to baseline methods.

**Questions To Authors:**

Why does TVP-loss work?

**Reasons To Accept:**

S1. A ​controlled experimental framework​ to study task vector emergence from scratch, addressing gaps in understanding how task information is encoded in pre-trained LLMs.

S2. A simple yet effective auxiliary training mechanism with a task vector prompting loss (TVP-loss),  that encourages task vectors to form at prescribed model layers, improving robustness to out-of-distribution (OOD) prompts.

S3. Expensive experiments across multiple tasks show its effectiveness.

**Reasons To Reject:**

W1. Although empirical results are strong, the paper lacks a ​theoretical explanation​ for why TVP-loss works.

W2. The synthetic nature of tasks limits conclusions about real-world applicability.

W3. TVP-loss adds computational overhead. Its scalability to larger models/architectures is also untested.

W4. The selection of some hyperparameters may require more explanation.

---

> ### Author Response · Authors · 2025-06-03
>
> We thank the reviewer for their thoughtful and constructive feedback. We are glad that our controlled framework, the simple and effective TVP-loss, and the extensive experiments across diverse tasks are recognized and appreciated. Below we address the reviewer’s concerns in detail.
>
> > Q1. Lack of theoretical explanation for why TVP-loss works. Why does TVP-loss work?
>
> Thank you for this important question. TVP-loss can be viewed as “working” in two ways: improving task vector identifiability (TVP performance) and improving OOD ICL performance.
>
> For the first, TVP-loss is explicitly designed to align intermediate representations with task-specific vectors. Minimizing this loss directly improves task vector quality, as supported by standard empirical risk minimization theory.
>
> For the second, we offer the following intuition: TVP-loss introduces a structural bias that encourages the model to encode task information consistently at specific layers. This regularization effect aligns with our empirical finding that task vectors naturally emerge in moderately sized models, suggesting that TVP-loss reinforces a behavior the model is already inclined toward under the proper capacity. A full theoretical explanation is left to future work.
>
> > Q2. Synthetic nature of tasks limits conclusions about real-world applicability
>
> Prior work [1] has demonstrated the emergence of task vectors in real-world LLMs under simple ICL tasks. Additionally, in Section 6.2, we show that applying TVP-loss during fine-tuning improves ICL performance on the MetaICL benchmark using a GPT2-Large (774M parameter) model. These results support our claim that task vectors can emerge in real-world LLM and that explicitly encouraging their formation enhances generalization in more realistic, natural language settings.
>
>     [1] Hendel, Roee, Mor Geva, and Amir Globerson. "In-context learning creates task vectors." arXiv preprint arXiv:2310.15916 (2023).
>
> > Q3. TVP-loss adds computational overhead; scalability to larger models is untested
>
> Thank you for raising this point. We acknowledge that TVP-loss introduces additional training-time overhead due to the extra forward pass and potential layer selection. In our experiments, this results in a modest 1.5 to 3x increase in training time for both small models and GPT2-Large (774M). Inference under ICL mode incurs no extra cost compared to the baseline.
>
> We note that no effort was made to optimize efficiency. We leave such improvements to future work, as our current focus is to characterize the emergence of task vectors and demonstrate their benefits.
>
> > Q4. Hyperparameter choices require more explanation
>
> Thank you for pointing this out. We will expand the description of our hyperparameter selection process in the Appendix. Specifically, for the synthetic tasks, we follow the previous setup [1,2]; for the formal language tasks, we follow the original paper [3, 4]'s setup, only adding a TVP-loss to the training procedure. For the MetaICL tasks, we also follow the experiment setup outlined in the original paper [5].
>
>     [1] Garg, Shivam, et al. "What can transformers learn in-context? a case study of simple function classes." Advances in Neural Information Processing Systems 35 (2022): 30583-30598.
>     [2] Lin, Ziqian, and Kangwook Lee. "Dual operating modes of in-context learning." Forty-first International Conference on Machine Learning. 2024.
>     [3] Xie, Sang Michael, et al. "An explanation of in-context learning as implicit bayesian inference." arXiv preprint arXiv:2111.02080 (2021).
>     [4] Akyürek, Ekin, et al. "In-context language learning: Architectures and algorithms." arXiv preprint arXiv:2401.12973 (2024).
>     [5] Min, Sewon, et al. "Metaicl: Learning to learn in context." arXiv preprint arXiv:2110.15943 (2021).

---

> > ### Comment · Reviewer_BR3X · 2025-06-11
> >
> > Thank you for your clarification. I maintain my score.

---

### Official Review · Reviewer_9mvp · 2025-05-20

**Rating:** 6
**Confidence:** 3
**Ethics Flag:** 1

**Summary:**

This study examines how task‑specific representations (“task vectors”) arise inside transformers during in‑context learning. Training models from scratch on synthetic data, the authors show that such vectors do emerge, but are often diffuse or weakly expressed. To concentrate and strengthen them at chosen layers, they introduce a task‑vector prompting loss (TVP‑loss) applied during training. TVP‑loss removes the need for post‑hoc searches for task‑correlated activations and leads to models with more robust, localized task vectors that generalize better across contexts.

**Reasons To Accept:**

1. The authors designed a good synthesized experiment that is small-scaled and easier to control. Thus they did a full set of ablation study to investigate the key factors for the natural emergence of task vectors.
2. This work proposed a simple yet effective way to strengthen the task encoding and improve the performance of the later task vector prompting.
3. The experiments are intensive and the results are informative.

**Reasons To Reject:**

The emergence of task vectors during training, as one of the major contributions, is not well supported by the results in Section 4.2.1. While the authors' conclusion is that task vectors are most distinct with **moderate** model depth, my interpretation can be that the "task vectors vanish when the model depth is large enough", which is the usual case for large language models. The condition for the emergence shown in Section 4.2.1 is so rigid that there is no specific implication on the LLM scenario.

---

> ### Author Response · Authors · 2025-06-03
>
> We thank the reviewer for recognizing our controlled synthetic setup and thorough ablations. We address your concern below.
>
> > "Task vectors vanish when the model depth is large enough"—disagrees with behavior in large language models.
>
> While standard LLMs are indeed deep, they are typically trained on datasets that are far larger and more diverse. In our synthetic experiments (Figure 3 and Figure 10), we observe that for linear regression task, the task vectors emerge most clearly in moderately capacity models--those with enough capacity to support structured task encoding, but not so much that task information can be handled without forming explicit representations. We conjecture that this regime--where model capacity is more closely aligned with task complexity--might share some properties with the conditions under which task vector emergences in large pretrained models [1]. We leave a deeper investigation of this connection to future work.
>
>     [1] Hendel, Roee, Mor Geva, and Amir Globerson. "In-context learning creates task vectors." arXiv preprint arXiv:2310.15916 (2023).

---

### Decision · Program_Chairs · 2025-07-08

**Decision:**

Accept

**Comment:**

The paper addresses the emergence of task vectors during training and suggests a loss that encourage such emergence. The reviewers appreciated the empirical setting, and the effectiveness of the new training approach.